# Attend to Anything: Foundation Model for Unified Human Attention Modeling

**Wenzhuo Zhao** [1]  **Ronghao Xian** [1]  **Keren Fu** [1 2]  **Qijun Zhao** [1 2]

## Abstract

Existing human attention (saliency) modeling methods persist as highly fragmented across modalities, scenes, and task formulations. Consequently, even with increasing model capacity and data scale, current models predominantly remain scene-dependent and task-specific, failing to practically generalize in real-world applications. To address the fundamental limitations, we present the Attend to Anything Model (AAM), a multi-modal foundation model that unifies attention modeling across various image, video, and audio-visual tasks and scenes. AAM reformulates attention as a cognitive entailment relationship organized in a general-to-specific hierarchy, implemented through language prompts with hierarchical embeddings in hyperbolic space. Furthermore, to unify static image and dynamic video attention, we adopt a fluid-dynamics perspective, formulating video-frame attention as a diffusive temporal evolution governed by the Fokker–Planck equation. Extensive experiments on 16 benchmarks demonstrate that AAM consistently outperforms state-of-the-art methods by an average of 6% across various scenarios, while achieving approximately a $4\times$ speedup in video inference. Overall, these results demonstrate that AAM provides a principled foundation for future research on attention and saliency-related tasks. The dataset and code will be available at https://github.com/wz-zhao/Attend-to-Anything.

## 1. Introduction

Human visual attention modeling (saliency prediction) aims to predict where humans look in visual stimuli and consti-

[1]College of Computer Science, Sichuan University, Chengdu, 610065, China [2]National Key Laboratory of Fundamental Science on Synthetic Vision, Sichuan University, Chengdu, 610065, China. Correspondence to: Keren Fu <fkrsuper@scu.edu.cn>.

*Proceedings of the 43rd International Conference on Machine Learning*, Seoul, South Korea. PMLR 306, 2026. Copyright 2026 by the author(s).

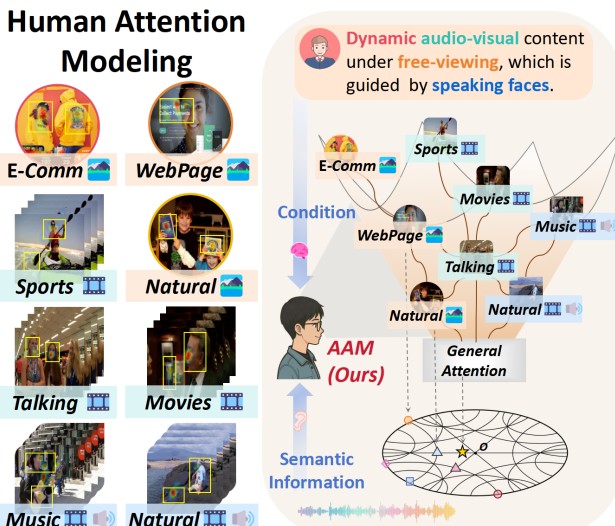

*Figure 1.* The Attend to Anything Model (AAM) unifies fragmented human attention tasks into a coherent foundation. By learning a general-to-specific hierarchy in hyperbolic space, AAM effectively models attention patterns across images, videos, and audio-visual scenarios, spanning from low-level visual saliency to complex, semantic-driven dynamic interactions.

tutes a fundamental component of multimedia understanding (Mishra et al., 2021), marketing analytics (Jiang et al., 2023), and robotic perception (Samani et al., 2023). Despite decades of progress across image, video, and audio-visual settings, existing attention models remain highly fragmented: different modalities, scenes, and task conditions are typically studied as **isolated problems** with dedicated architectures and training protocols (Zhou et al., 2023; Tang et al., 2025). This fragmentation stands in stark contrast to recent advances in computer vision, where unified foundation models have demonstrated strong generalization across data domains and tasks. To date, attention modeling still lacks a unified **foundation model** capable of generalizing across modalities, scenes, and tasks, which severely limits its generalization ability and real-world applicability. The persistent gap in cross-dataset generalization, despite increasing model capacity and data scale (Kümmerer et al., 2025), points to a deeper issue: the current formulation of attention as disjoint tasks fails to capture its underlying unified cognitive process. The absence of a unified foundation model in attention modeling can be traced to **two fundamental challenges** rooted in current problem formulations:

*I) Cross-Scene generalization.* From a neuroscientific perspective (Rao & Ballard, 1999), human visual attention is **hierarchically** modulated by cognitive context and tasks, a property widely acknowledged in attention modeling research (Yarbus, 2013; Lou et al., 2022). However, existing methods (Droste et al., 2020; Chen et al., 2023a) predominantly attribute condition-dependent variations to dataset-specific statistical biases, thereby collapsing the hierarchical modulation (**from general to specific**) into dataset-specific differences. Under this paradigm, attention models are typically trained and evaluated on individual datasets (Xie et al., 2024; Jin et al., 2025), leading to a performance plateau on existing benchmarks (Kümmerer & Bethge, 2023). Recent studies further reveal a substantial performance drop (around **40%**) when models trained on one dataset are applied to another (Kümmerer et al., 2025), a discrepancy that cannot be resolved simply by scaling training data.

Although several works attempt to alleviate this issue via joint training across multiple datasets (Droste et al., 2020; Hosseini et al., 2025b), they rely on dataset-specific priors or isolated parameters (e.g., normalization statistics or Gaussian maps), fundamentally limiting their ability to generalize beyond the statistics of observed datasets.

*II) Cross-Task modeling.* A second challenge arises from the **heterogeneous formulation** of attention modeling across modalities and temporal scales. Image and video attention have long been treated as distinct tasks, despite arising from the same underlying attentional mechanisms. Existing video models operate on **fixed-window clips** and encode temporal information through optical flow (Lai et al., 2019) or spatiotemporal convolutions and transformers (Zhou et al., 2023), only outputting the final frame. Such formulations impose task boundaries, restrict frame-wise inference, and incur substantial computational overhead, limiting modeling flexibility and inference efficiency.

To address these challenges, we propose the **Attend to Anything Model (AAM)**, the first unified multi-modal foundation model for attention modeling across images, videos, and audio-visual scenarios. ❶ To tackle **cross-scene generalization**, we structure the **cognitive evolution** from general priors to specific tasks as an asymmetric hierarchical implication (Fig. 1). Since hyperbolic space naturally supports hierarchical structures (Li et al., 2025a; Liu et al., 2025), we model attention differences as **entailment relations** induced by text prompts in **hyperbolic** space. Notably, compared to existing methods based on parameter isolation, AAM introduces a cognitively motivated paradigm for modeling attention variation, offering theoretical and empirical insights for the future development of holistic visual perception systems. ❷ To tackle **cross-task modeling**, we introduce the Fokker–Planck Dynamics Module (FPD), which models video attention as the **fluid dynamics** of static attention

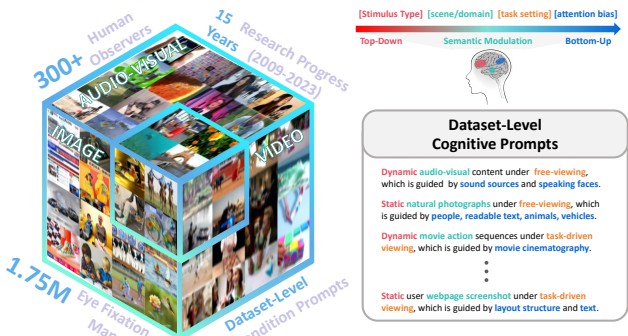

*Figure 2.* Attention-1.75M surpasses existing datasets in both scale and diversity, and is equipped with dataset-level texts derived from the dataset acquisition protocols (Appendix B for details).

over a spatiotemporal manifold. FPD models attention as the continuous transport governed by the **Fokker–Planck** equation, decomposing the evolution process into advection and diffusion. This physics-informed formulation unifies static and dynamic attention, enabling efficient frame-wise inference. ❸ To support the foundation model training, we curate **Attention-1.75M**, a standardized corpus unifying over 1.75M fixation instances across images, videos, and audio-visual scenarios equipped with dataset-level texts. Our principal contributions are summarized as follows:

❶ *Problem Identification.* We identify a **fundamental mismatch** between unified human attentional mechanisms and existing fragmented attention modeling formulations, revealing that the absence of a unified foundation model severely limits generalization and real-world applicability.

❷ *Paradigm Reformulation.* We propose Attend to Anything Model (AAM), the first unified multi-modal foundation model for attention modeling across image, video, and audio-visual scenarios. AAM achieves cross-scene generalization via hierarchical entailment in hyperbolic space and resolves static–dynamic task incompatibility through Fokker–Planck physical temporal dynamics, unifying image and video attention in a continuous framework.

❸ *Experimental Validation.* We evaluate AAM on 16 benchmarks, demonstrating consistent state-of-the-art (SOTA) performance with an average improvement of **6%** and approximately a **4×** speedup in video inference, supported by a large-scale training corpus of **1.75M** instances.

❹ *Conceptual Insights.* Extensive inductive experiments demonstrate that AAM introduces a cognitively aligned paradigm for modeling attentional differences, contributing theoretical understanding and empirical evidence to the future development of holistic visual perception systems.

## 2. Related Works

**Evolution of Attention Modeling.** The field has evolved from heuristic contrast features (Riche et al., 2013) to deep

representation learning across three tracks. Image modeling employs CNN and Transformer backbones (Huang et al., 2015; Lou et al., 2022) to learn distribution mappings, but typically ignores the hierarchical nature of cognitive tasks. Temporal modeling extends these to videos via motion cues (Jain et al., 2021) or spatiotemporal encoders such as 3D-CNNs and Video Swin Transformers (Bellitto et al., 2021; Jin et al., 2025), which are often constrained by fixed-window and high complexity. Audio-visual modeling integrates auditory streams through sound localization or modality-specific fusion branches (Aytar et al., 2016; Xiong et al., 2024). Despite recent joint-training attempts (Droste et al., 2020; Hosseini et al., 2025b; Kümmerer et al., 2025), existing methods still rely on parameter isolation or specialized modules to handle heterogeneous data. In contrast, AAM provides a unified paradigm by modeling attention in hyperbolic space and bridging static-dynamic transitions through a continuous fluid-dynamics formulation.

**Biological and Cognitive Foundations of Attention.** Existing studies (Jiang et al., 2024) in neuroscience and cognitive science provide important theoretical support for modeling attention in a cognitively aligned manner. First, the hierarchical organization of the visual cortex, from low-level edge responses in V1 to high-level semantic representations in IT, suggests that visual perception is inherently structured across multiple levels of abstraction. Prior work has shown that visual hypercolumns can be naturally modeled with hyperbolic geometry (Chossat & Faugeras, 2009), and that spiking activity patterns in V1/V2 exhibit intrinsic hyperbolic structure (Guidolin et al., 2022). These findings motivate the use of hyperbolic space to capture the hierarchical attention. Second, the drift-diffusion model (DDM) has been widely used to explain perceptual decision-making and saccadic latency in cognitive science (Ratcliff, 1978; Bogacz et al., 2006). Its macroscopic dynamics are closely related to the Fokker–Planck equation, which describes the evolution of probability density over time. This connection provides a biologically grounded interpretation of attention shifts: drift captures top-down task-driven bias, diffusion reflects bottom-up visual exploration, and the concentration of probability mass corresponds to decision-boundary crossing (Shinn et al., 2020). Together, these biological and cognitive studies motivate AAM as a formulation that bridges static geometry and temporal attention dynamics.

## 3. Methodology

### 3.1. Overview of AAM

In this section, we present an overview of the proposed AAM, which represents human attention as a shared latent process across modalities, scenes, and time by modeling hierarchical semantic specialization and temporal dynamics, as shown in Fig. 3. **Visual** inputs are encoded by a frozen self-supervised backbone (DINOv3 (Siméoni et al., 2025)) with LoRA for adaptation to attention modeling (see Appendix A for detailed hyperparameters and configurations). **Text** prompts and **audio** signals are encoded using frozen CLIP (Radford et al., 2021) and Wav2CLIP (Wu et al., 2022) encoders, respectively, with audio mapped into the visual semantic space. Audio-visual fusion is performed through a relevance-gated cross-attention mechanism, ensuring that audio cues contribute only when semantically aligned (Appendix E.2). For video inputs, frame-wise attention representations are refined by a Fokker–Planck Dynamics (FPD) module that models attention evolution over a spatiotemporal manifold (Sec. 3.4). Visual and textual representations are lifted into hyperbolic space via hierarchical entailment learning for explicit hierarchy enforcement (Sec. 3.2). Finally, a geometry-aware hyperbolic decoder projects structured representations back to Euclidean space to generate spatial attention maps (Sec. 3.3).

### 3.2. Hierarchical Human Attention Modeling in Hyperbolic Space

#### 3.2.1. PRELIMINARIES

Hyperbolic geometry, a non-Euclidean geometry characterized by constant negative curvature and exponential volume growth, naturally encodes the degree of semantic specialization through the distance from the origin and represents the scope of refinement via angular regions. This geometric structure makes it an ideal choice for learning representations of data with inherent hierarchical structures (Krioukov et al., 2010; Sarkar, 2011; Nickel & Kiela, 2017). Specifically, the Lorentz model $\mathbb{L}^n_\kappa$ with curvature $-\kappa \in \mathbb{R}$ is defined as an $n$-dimensional manifold represented as the upper sheet of a two-sheeted hyperboloid in $(n + 1)$-dimensional Minkowski spacetime, which is described as

$$\mathbb{L}^n_\kappa = \left\{ \mathbf{z} \in \mathbb{R}^{n+1} : \langle \mathbf{z}, \mathbf{z} \rangle_{\mathbb{L}} = -\frac{1}{\kappa}, \ \ \mathbf{z}_0 = \sqrt{\frac{1}{\kappa} + \|\tilde{\mathbf{z}}\|^2_2} \right\},$$
(1)

where $\langle ., . \rangle_{\mathbb{L}}$ denotes the Lorentzian inner product for $\mathbf{z}, \mathbf{z}' \in \mathbb{L}^n_\kappa$, defined as:

$$\langle \mathbf{z}, \mathbf{z}' \rangle_{\mathbb{L}} = -\mathbf{z}_0 \mathbf{z}'_0 + \langle \tilde{\mathbf{z}}, \tilde{\mathbf{z}}' \rangle_{\mathbb{E}},$$
(2)

with $\langle ., . \rangle_{\mathbb{E}}$ denoting the Euclidean inner product. For each vector $\mathbf{z}$, the first dimension is taken as the *time*-axis, denoted $\mathbf{z}_0$, and the remaining $n$ dimensions as the *spatial*-coordinates, denoted $\tilde{\mathbf{z}} \in \mathbb{R}^n$. The Lorentzian distance between two points in $\mathbb{L}^n_\kappa$, is the length of the shortest path (*geodesic*), which can be computed as:

$$d_{\mathbb{L}}(\mathbf{z}, \mathbf{z}') = \sqrt{\frac{1}{\kappa}} \cosh^{-1}(-\kappa \langle \mathbf{z}, \mathbf{z}' \rangle_{\mathbb{L}}), \quad \mathbf{z}, \mathbf{z}' \in \mathbb{L}^n_\kappa, \ (3)$$

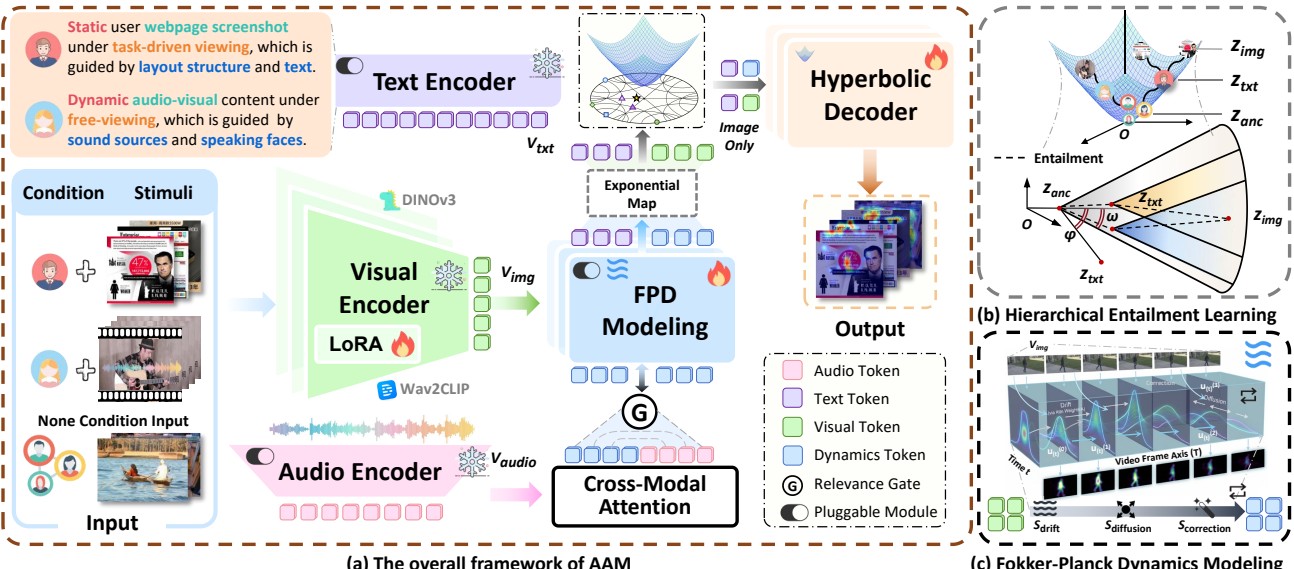

*Figure 3.* (a) Overview of AAM. (b) The general-specific attention ordering between $(\mathbf{z}_{\text{anc}}, \mathbf{z}_{\text{txt}})$, $(\mathbf{z}_{\text{txt}}, \mathbf{z}_{\text{img}})$ is enforced in hyperbolic space using entailment cones. The external angle $\phi$ of a specific condition $(\mathbf{z}_{\text{txt}})$ is pushed to be within the aperture threshold $\eta\omega$ of the general attention $(\mathbf{z}_{\text{anc}})$. (c) The Fokker–Planck Dynamics (FPD) Modeling illustrates the drift, diffusion, and correction processes used to model the transition of attention over the video frame axis.

which inducing Lorentzian norm $\|\mathbf{z}\|_{\mathbb{L}} = \langle \mathbf{z}, \mathbf{z} \rangle_{\mathbb{L}}$. Based on this geometry, any vector $\mathbf{v} \in T_{\mathbf{z}} \mathbb{L}_{\kappa}^n$ in the tangent space can be projected onto the hyperboloid using the exponential map (Khrulkov et al., 2020):

$$\exp_{\mathbf{z}}^{\kappa}(\mathbf{v}) = \cosh\left(\sqrt{\kappa}\|\mathbf{v}\|_{\mathbb{L}}\right)\mathbf{z} + \frac{\sinh\left(\sqrt{\kappa}\|\mathbf{v}\|_{\mathbb{L}}\right)}{\sqrt{\kappa}\|\mathbf{v}\|_{\mathbb{L}}}\mathbf{v}. \quad (4)$$

### 3.2.2. HIERARCHICAL ENTAILMENT LEARNING

We model human visual attention as a hierarchical process. Specifically, attention is refined from general attention to specific visual attention distribution as conditions are imposed. To formalize these hierarchical entailment relations, we introduce a **partial order relation** in hyperbolic space:

$$\mathbf{z}_{\text{img}} \preceq \mathbf{z}_{\text{txt}} \preceq \mathbf{z}_{\text{anc}}, \quad (5)$$

where $\mathbf{z}_{\text{anc}}$ and $\mathbf{z}_{\text{txt}}$ denote the general attention and text prompts, respectively, with $\mathbf{z}_{\text{img}}$ serving as their specific instantiation within the visual input. We map the text embedding $\mathbf{v}_{\text{txt}}$ generated by the text encoder into the Lorentz manifold via the exponential map defined at the origin:

$$\mathbf{z}_{\text{txt}} = \exp_{\mathbf{o}}^{\kappa}(\mathbf{v}_{\text{txt}}) \in \mathbb{L}_{\kappa}^n. \quad (6)$$

Similarly, we map the visual feature embedding $\mathbf{v}_{\text{img}}$ into the Lorentz manifold to obtain its hyperbolic representation $\mathbf{z}_{\text{img}}$, and introduce a learnable general attention anchor $\mathbf{z}_{\text{anc}}$.

Based on these hyperbolic representations, we employ hyperbolic entailment cones (Ganea et al., 2018; Pal et al., 2025) to transform hierarchical attention entailment relations into optimizable geometric constraints. As illustrated

in Fig. 3 (b), entailment cones define a region $\mathfrak{R}_{\mathbf{z}_{\text{anc}}}$ for every possible point $\mathbf{z}_{\text{anc}}$ in the space such that all points $\mathbf{z}_{\text{txt}} \in \mathfrak{R}_{\mathbf{z}_{\text{anc}}}$ are semantically linked to $\mathbf{z}_{\text{anc}}$ as its child concepts. As such, points in $\mathfrak{R}_{\mathbf{z}_{\text{anc}}}$ are expected to contain specific condition for the general concept ($\mathbf{z}_{\text{txt}} \preceq \mathbf{z}_{\text{anc}}$). The half-aperture of these conical regions is formulated by (Le et al., 2019; Desai et al., 2023) as: $\omega(\mathbf{z}) = \sin^{-1}\left(\frac{2K}{\sqrt{\kappa}\|\tilde{\mathbf{z}}\|}\right)$.

To learn partial orders in the Lorentz space, we employ a thresholded entailment loss based on angular residuals (Le et al., 2019; Desai et al., 2023):

$$\mathcal{L}_{\text{ent}}^*(\mathbf{z}_{\text{anc}}, \mathbf{z}_{\text{txt}}) = \max(0, \; \phi(\mathbf{z}_{\text{anc}}, \mathbf{z}_{\text{txt}}) - \eta\,\omega(\mathbf{z}_{\text{anc}})), \quad (7)$$

where $\eta$ is a threshold scaling factor used to adjust the tightness of the entailment constraint, $\phi(\mathbf{z}, \mathbf{z}')$ denotes the exterior angle of the child node deviating from the boundary of the parent entailment cone:

$$\phi(\mathbf{z}, \mathbf{z}') = \cos^{-1}\left(\frac{z_0 + z_0'\,\kappa\langle \mathbf{z}, \mathbf{z}'\rangle_{\mathbb{L}}}{\|\tilde{\mathbf{z}}'\|\sqrt{(\kappa\langle \mathbf{z}, \mathbf{z}'\rangle_{\mathbb{L}})^2 - 1}}\right). \quad (8)$$

Hence, the hierarchical attention entailment (HAE) loss of human attention would comprise both anchor-text conditional entailments and text-image entailments as:

$$\mathcal{L}_{\text{HAE}} = \mathcal{L}_{\text{ent}}^*(\mathbf{z}_{\text{anc}}, \mathbf{z}_{\text{txt}}) + \lambda\,\mathcal{L}_{\text{ent}}^*(\mathbf{z}_{\text{txt}}, \mathbf{z}_{\text{img}}), \quad (9)$$

The final loss $\mathcal{L}_{\text{total}}$ (Droste et al., 2020) is composed of task and HAE loss:

$$\mathcal{L}_{\text{total}} = \mathcal{L}_{\text{KLD}} - \mathcal{L}_{\text{CC}} - \mathcal{L}_{\text{SIM}} + \mathcal{L}_{\text{HAE}}. \quad (10)$$

$\mathcal{L}_{\text{KLD}},\mathcal{L}_{\text{CC}}$ ,$\mathcal{L}_{\text{SIM}}$ represent the task losses for attention modeling, as detailed in Appendix A.1

### 3.3. Decoding Attention from Hyperbolic Manifolds

Building on the hierarchical entailment constraints regarding text specialization depth and cone positioning, the hyperbolic decoder employs scale modulation and spatial focusing to map hyperbolic geometry into Euclidean attention. The visual features $\mathbf{X} \in \mathbb{R}^{C \times H \times W}$ are thus adaptively modulated by $\mathbf{z}_{\text{img}}, \mathbf{z}_{\text{txt}}$ (Architecture details in Appendix 1).

❶ **Specialization depth-driven scale modulation.** Given that geodesic distance in hyperbolic space characterizes semantic specialization, we define the *specialization depth* of the condition as $r_{\text{txt}} = d_{\mathbb{L}}(\mathbf{z}_{\text{txt}}, \mathbf{o})$, which regulates the relative focus of the decoder between global structure and fine-grained local details. We introduce a set of operators $\{\mathcal{S}_k\}_{k=1}^{K}$ with $K$ levels alongside anchors $\boldsymbol{\mu}_k \in \mathbb{L}_\kappa^n$ to perform conditional weighted fusion of multi-scale features. The scale weights are determined by the hyperbolic distances between the text condition and scale anchor:

$$w_k = \text{softmax}_k\left(-d_{\mathbb{L}}(\mathbf{z}_{\text{txt}}, \boldsymbol{\mu}_k)\right). \qquad (11)$$

The overall modulation intensity is governed by a monotonic function of the specialization depth, $\alpha(r_{\text{txt}}) = \text{softplus}(r_{\text{txt}})$, while the final scale response is computed as a weighted combination of the operators:

$$\mathbf{X}_s = \sum_{k=1}^{K} w_k\, \mathcal{S}_k(\mathbf{X}). \qquad (12)$$

❷ **Relative geodesic direction-driven spatial focusing.** To characterize the relative semantic relationship between the condition and the visual instance, we define the relative geodesic direction in the tangent space at the origin:

$$\Delta = \log_{\mathbf{o}}^{\kappa}(\mathbf{z}_{\text{img}}) - \log_{\mathbf{o}}^{\kappa}(\mathbf{z}_{\text{txt}}). \qquad (13)$$

$\Delta$ captures the semantic deviation direction of the visual instance relative to the condition center, thereby providing semantic guidance for pixel-level spatial focusing. The aperture of the entailment cone corresponding to the text condition, denoted as $\omega(\mathbf{z}_{\text{txt}})$, reflects its degree of semantic generalization and regulates the intensity of spatial focusing. We define the focusing temperature as $\beta = \beta_0(\omega(\mathbf{z}_{\text{txt}})+\varepsilon)$, where $\beta_0$ is a temperature scaling hyperparameter. A larger cone aperture corresponds to a more generalized semantic condition, inducing a broader spatial attention pattern. Let $\mathbf{u}_{i,j}$ denote the channel-wise feature vector of $\mathbf{X}$ at position $(i,j)$; the spatial weight is defined by its consistency with the relative geodesic direction $\Delta$:

$$m_{i,j} = \text{softmax}_{i,j}\left(\frac{\langle\mathbf{u}_{i,j}, \Delta\rangle}{\|\mathbf{u}_{i,j}\|\,\|\Delta\| + \varepsilon} \cdot \frac{1}{\beta}\right). \qquad (14)$$

❸ **Joint decoding.** Scale selection and spatial focusing are integrated via a unified residual formulation:

$$\mathbf{X}' = \mathbf{X} + \alpha(r_{\text{txt}})\Big(\mathbf{M} \odot \mathbf{X}_s\Big), \qquad (15)$$

where $\mathbf{M} = \{m_{i,j}\}$ denotes the attention map induced by the relative geodesic direction. This residual fusion ensures that the hyperbolic entailment structure is consistently preserved within the pixel-wise attention distribution.

### 3.4. Fokker–Planck Dynamics (FPD) Modeling

For video sequences of **arbitrary length**, let $u_t^{\text{obs}}$ denote the attention distribution derived from the visual encoder output $\mathbf{v}_{\text{img}}$, defined on the discrete spatial domain $\Omega = \{1, \ldots, H\} \times \{1, \ldots, W\}$ and subject to a normalization constraint. We introduce the Fokker–Planck (FP) dynamics equation to model the temporal evolution of attention:

$$\frac{\partial u}{\partial t} = \underbrace{-\nabla_\tau \cdot (\mathbf{v}u)}_{\mathcal{A}_{\text{drift}}} + \underbrace{\nabla_\tau \cdot (D\nabla_\tau u)}_{\mathcal{A}_{\text{diffusion}}} + \underbrace{\lambda(u^{\text{obs}} - u)}_{\mathcal{A}_{\text{correction}}}, \quad (16)$$

where the operators $\nabla_\tau$ are defined along the temporal axis, characterizing attention drifting, smoothing, and correction, respectively. The parameter $\lambda$ balances the dynamic prediction with the initial information. We employ the Lie–Trotter operator splitting scheme to discretize the FP dynamics into sequential sub-operator updates within a time step $\Delta t$:

$$u(t+\Delta t) \approx (\mathcal{S}_{\text{drift}} \circ \mathcal{S}_{\text{diffusion}} \circ \mathcal{S}_{\text{correction}} \circ \mathcal{P}_{\text{proj}})[u(t)]. \quad (17)$$

We denote by $u_t^{(k)}$ the intermediate state after applying the $k$-th sub-operator, with $u_t^{(0)} \equiv u_t$.

#### 3.4.1. DRIFT EVOLUTION OPERATOR: $\mathcal{S}_{\text{DRIFT}}$

The drift equation is given by $\frac{\partial u}{\partial t} = -\nabla_\tau \cdot (\mathbf{v}u)$. To simulate this physical process within a discrete feature space, we utilize bidirectional temporal self-attention to parameterize the drift propagator $A_x$. We define the discrete transition kernel $A_x(t \leftarrow t')$ from source time $t'$ to target time $t$ as:

$$A_x^{(h)}(t \leftarrow t') = \frac{\exp\left(\langle q_t^{(h)}(x), k_{t'}^{(h)}(x)\rangle/(\sqrt{d}\beta))\right)}{\sum_{\tau'=1}^{T} \exp\left(\langle q_t^{(h)}(x), k_{\tau'}^{(h)}(x)\rangle/(\sqrt{d}\beta)\right)}. \tag{18}$$

Here, the transition kernel $A_x$ is normalized along the temporal axis to facilitate cross-time information aggregation, constituting a Markov transition (Detailed specifications are provided in Appendix D.2). Following the Lagrangian transport, $\tilde{u}_t$ denotes the aggregated state derived from information integration across the full temporal domain:

$$\tilde{u}_t(x) = \sum_{t'=1}^{T} A_x(t \leftarrow t')u_{t'}(x). \qquad (19)$$

Subsequently, the drift operator $\mathcal{S}_{\text{drift}}$ adopts a residual Euler update scheme to evolve the state $u_t^{(0)}$ into the first-stage intermediate state $u_t^{(1)}$:

$$u_t^{(1)}(x) = u_t^{(0)}(x) + \Delta t \cdot \left( \tilde{u}_t(x) - u_t^{(0)}(x) \right). \quad (20)$$

The full-temporal attention mechanism enables cross-time information backflow, effectively mitigating the issue of insufficient motion cues in the early stages.

### 3.4.2. DIFFUSION OPERATOR: $\mathcal{S}_{\text{DIFFUSION}}$

To regularize the high-frequency noise introduced during the drift process, we employ a second-order central finite difference approximation based on the intermediate state $u_t^{(1)}$ to compute the temporal diffusion term:

$$u_t^{(2)}(x) = u_t^{(1)}(x) + \nu_t(x)\Delta t \frac{u_{t-1}^{(1)}(x) - 2u_t^{(1)}(x) + u_{t+1}^{(1)}(x)}{(\Delta t)^2}. \quad (21)$$

This second-order central difference approximation performs temporal smoothing, where the learnable intensity $\nu_t(x)$ adapts by reducing diffusion in dynamic regions to preserve edges and increasing it in static areas for stability.

### 3.4.3. CORRECTION OPERATOR: $\mathcal{S}_{\text{CORRECTION}}$

Finally, we employ a relaxed Euler scheme to refine the prediction based on the original state:

$$u_t^{(3)}(x) = (1 - \lambda_t(x))u_t^{(2)}(x) + \lambda_t(x)u_t^{\text{obs}}(x). \quad (22)$$

The coefficient $\lambda_t(x) \in [0, 1]$ is a learnable gating parameter activated via a Sigmoid function. This term functions analogously to a Kalman gain, dynamically correcting between the dynamic prediction $u_t^{(2)}$ and the global static features $u_t^{\text{obs}}$, thereby suppressing error accumulation. To address simplex deviations caused by the correction term, we project $u_t^{(3)}$ via $\mathcal{P}_{\text{proj}}$ to ensure numerical stability for the next iteration.

$$u_t(x) \leftarrow \frac{u_t^{(3)}(x) + \epsilon}{\sum_{x' \in \Omega} (u_t^{(3)}(x') + \epsilon)}. \quad (23)$$

## 4. Experiments

### 4.1. Experimental Setup

**Datasets.** AAM is trained on Attention-1.75M, a standardized corpus unifying over 1.75M fixation instances across 8 image, 4 video, and 6 audio-visual datasets. Dataset specifications and textual conditions are provided in Appendix B. Covering diverse scenarios, Attention-1.75M substantially exceeds the training scale of existing methods.

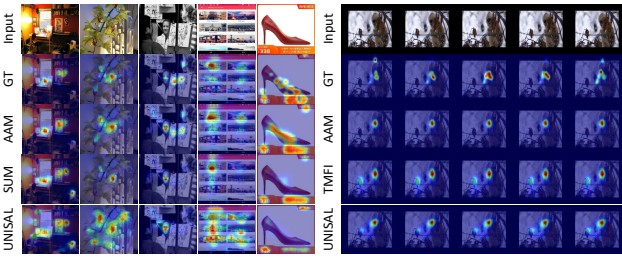

(a) Image Attention Modeling  (b) Video Attention Modeling

*Figure 4.* Visual comparison against SOTA methods.

**Implementation Details.** AAM is implemented with the NPU-PyTorch CANN on four Ascend Snt9B3 NPUs. All inputs are resized to $448 \times 448$. We employ a phased training strategy: AAM is first trained on image and video data (with a sampling ratio of 4:6), during which the general attention anchor $z_{\text{anc}}$ is **warm-started** by free-viewing datasets. Audio-visual data is then added after 10 epochs (Detailed configuration in Appendix 6).

**Baselines.** We compare our foundation model with state-of-the-art attention modeling methods including (i) scene-dependent, task-specific models and (ii) jointly trained models with parameter isolation across partial datasets, as discussed in Section 1 (e.g., SUM and UNISAL).

### 4.2. Performance Analysis

Extensive experiments across 16 datasets demonstrate that AAM consistently outperforms SOTA methods using a single model. As in prior work (Hosseini et al., 2025b), we use the evaluation metrics AUC-Judd (AUC), Similarity Metric (SIM), Linear Correlation Coefficient (CC), and Normalized Scanpath Saliency (NSS). Specifically, averaging improvements across all evaluation metrics, we achieve average metric gains of 5.2%, 5.8% (including LEDOV in Appendix 21), and 6.0% on image (Table 1), audio-visual (Table 2), and video (Table 3) tasks, respectively (see Appendix F.4 for detailed comparison). Notably, diverging from existing methods that rely on computationally intensive multi-to-single-frame paradigms, AAM employs an efficient frame-wise prediction via FPD. This architectural distinction not only secures robust performance within a unified framework but also eliminates inherent redundancy. Consequently, AAM achieves an approximate $4\times$ inference speedup compared to fixed-window methods (Table 4), with only 21M trainable parameters (324M total). Qualitative visual comparisons are presented in Fig. 4.

### 4.3. Ablation Study

#### 4.3.1. ANALYSIS OF JOINT TRAINING

As shown in Fig. 5, all results are averaged over datasets. Under our hierarchical conditional modeling setting, we

*Table 1.* Quantitative comparison on **image** datasets. The best results are shown in red, and the second best in blue.

| 🖼 Image Attention Modeling | | | | | | | | |
|---|---|---|---|---|---|---|---|---|
| **Method** | **CC↑** | **KLD↓** | **AUC↑** | **SIM↑** | **Method** | **CC↑** | **KLD↓** | **AUC↑** | **SIM↑** |
| *Dataset: MIT1003 (Natural)* | | | | | *Dataset: U-EYE (Web page)* | | | | |
| UNISAL (Droste et al., 2020) | 0.734 | 1.014 | 0.902 | 0.597 | TransalNet (Lou et al., 2022) | 0.696 | 0.616 | 0.839 | 0.598 |
| TransalNet (Lou et al., 2022) | 0.722 | 0.660 | 0.903 | 0.592 | UMSI++ (Jiang et al., 2023) | 0.670 | 0.860 | 0.830 | 0.580 |
| SUM (Hosseini et al., 2025b) | 0.768 | 0.563 | 0.913 | 0.630 | SUM (Hosseini et al., 2025b) | 0.731 | 0.544 | 0.846 | 0.630 |
| AAM (Ours) | 0.831 | 0.446 | 0.923 | 0.674 | AAM (Ours) | 0.743 | 0.524 | 0.847 | 0.635 |
| *Dataset: CAT2000 (Natural)* | | | | | *Dataset: SalECI (E-Commercial)* | | | | |
| UNISAL (Droste et al., 2020) | 0.842 | 0.530 | 0.876 | 0.721 | EML-NET (Jiang et al., 2023) | 0.510 | 1.220 | 0.807 | 0.536 |
| TransalNet (Lou et al., 2022) | 0.877 | 0.287 | 0.882 | 0.744 | Hosseini (Hosseini et al., 2025a) | 0.750 | 0.578 | 0.892 | 0.645 |
| SUM (Hosseini et al., 2025b) | 0.882 | 0.270 | 0.888 | 0.754 | SUM (Hosseini et al., 2025b) | 0.789 | 0.473 | 0.899 | 0.680 |
| AAM (Ours) | 0.906 | 0.235 | 0.890 | 0.769 | AAM (Ours) | 0.797 | 0.450 | 0.899 | 0.678 |
| *Dataset: SALICON (Natural)* | | | | | *Dataset: OSIE (Natural)* | | | | |
| TransalNet (Lou et al., 2022) | 0.890 | 0.220 | 0.867 | 0.783 | TransalNet (Jiang et al., 2023) | 0.791 | 0.667 | 0.923 | 0.651 |
| Temp-Sal (Aydemir et al., 2023) | 0.911 | 0.195 | 0.869 | 0.800 | UniAR (Li et al., 2024) | 0.754 | 0.547 | 0.867 | 0.647 |
| SUM (Hosseini et al., 2025b) | 0.909 | 0.192 | 0.876 | 0.804 | SUM (Hosseini et al., 2025b) | 0.861 | 0.340 | 0.924 | 0.727 |
| AAM (Ours) | 0.925 | 0.163 | 0.876 | 0.819 | AAM (Ours) | 0.901 | 0.243 | 0.933 | 0.760 |

*Table 2.* Quantitative comparison on **audio-visual** datasets. The best results are shown in red, and the second best in blue.

| 🔊⊞ Audio-Video Attention Modeling | | | | | | | | | | | | | | | |
|---|---|---|---|---|---|---|---|---|---|---|---|---|---|---|---|
| **Method** | **DIEM** | | | **ETMD** | | | **SumMe** | | | **Coutrot1** | | | **Coutrot2** | | |
| | **CC↑** | **NSS↑** | **AUC↑** | **CC↑** | **NSS↑** | **AUC↑** | **CC↑** | **NSS↑** | **AUC↑** | **CC↑** | **NSS↑** | **AUC↑** | **CC↑** | **NSS↑** | **AUC↑** |
| CASP (Xiong et al., 2023) | 0.655 | 2.61 | 0.906 | 0.620 | 3.34 | 0.940 | 0.499 | 2.60 | 0.907 | 0.561 | 2.65 | 0.889 | 0.788 | 6.34 | 0.963 |
| DAVS (Zhu et al., 2024) | 0.580 | 2.29 | 0.884 | 0.600 | 2.96 | 0.932 | 0.423 | 2.29 | 0.889 | 0.482 | 2.19 | 0.869 | 0.734 | 4.98 | 0.960 |
| MSPI (Xie et al., 2024) | 0.653 | 2.62 | 0.907 | 0.601 | 3.24 | 0.937 | 0.482 | 2.49 | 0.901 | 0.567 | 2.76 | 0.895 | 0.783 | 6.28 | 0.963 |
| TAVDiff (Yu et al., 2025) | 0.670 | 2.75 | 0.909 | 0.613 | 3.15 | 0.937 | 0.500 | 2.51 | 0.904 | 0.607 | 2.85 | 0.892 | 0.798 | 6.52 | 0.963 |
| AAM (Ours) | 0.710 | 2.88 | 0.919 | 0.655 | 3.66 | 0.945 | 0.550 | 2.90 | 0.920 | 0.626 | 3.22 | 0.911 | 0.887 | 7.46 | 0.971 |

conduct a systematic ablation study to evaluate the effect of unified multimodal joint training. For **image** joint training (A), A1 evaluates cross-dataset generalization trained on single-dataset. A2 is the in-domain single-dataset baseline, where the model is trained and evaluated separately on each dataset. A3 performs full images joint training, and A4 further incorporates video data for cross-modal joint training. For **audio-visual** training (B), B1 is the in-domain single-dataset baseline, B2 adds video data, and B3 applies full multimodal unified training, leading to consistent improvements. For **video** joint training (C), C1 is the in-domain single-dataset baseline, C2 adds video data, C3 combines video and image data, and C4 trains on all data jointly, yielding stable gains across benchmarks. Overall, these results support our hypothesis that unified multimodal training enhances generalization under hierarchical attention modeling.

### 4.3.2. COMPONENT ABLATION

We conduct a systematic ablation study on the proposed components. For the backbone ablation (D), we compare different visual encoder configurations, including DINOv3 (D1–D3, small→large) and SAM2 (Ravi et al., 2025) (D4 base, D5 large). The results indicate that stronger self-supervised visual representations further improve overall performance. For the temporal modeling ablation (E), the model progresses from removing the temporal module entirely (E1), to adopting a standard time-dimension self-

attention (Chen et al., 2025) (E2), and finally to our proposed FPD temporal module (E3/E4), which uses 16-frame and 32-frame clips as input, respectively. The full temporal design achieves the best performance, supporting our motivation to model video attention as a continuous evolution over a spatiotemporal manifold. For the hyperbolic representation ablation (F), performance consistently improves from removing hyperbolic learning (F1), to introducing a hyperbolic loss constraint (F2), and further to incorporating a hyperbolic decoder enhancement (F3), with more pronounced gains in complex scenarios exhibiting stronger hierarchical structure. In contrast, comparisons with the joint-training settings in (A) suggest that naive direct joint training may introduce domain conflicts and lead to degraded performance.

### 4.3.3. INDUCTIVE ANALYSIS

To analyze whether AAM effectively models the hierarchical attention of cognitive modulation, we conduct a series of ablation studies under varying semantic conditions, as shown in Fig. 6 (see Appendix F for detailed setups). We assess the cognitive regulation from three perspectives:

❶ **Condition swap.** We evaluate three configurations: Correct (task-aligned), Generic (general attention), and Wrong (mismatched tasks). We observe a strict performance hierarchy: *Correct > Generic > Wrong*, despite identical visual inputs. Such sensitivity to condition swapping demon-

*Table 3.* Quantitative comparison on **video** datasets. The best results are shown in red, and the second best in blue.

| | ⊞ Video Attention Modeling | | | | | | | | | | |
|---|---|---|---|---|---|---|---|---|---|---|---|
| **Method** | **DHF1K (Natural)** | | | | **Hollywood2 (Movies)** | | | | **UCF (Sports)** | | |
| | AUC↑ | SIM↑ | CC↑ | NSS↑ | AUC↑ | SIM↑ | CC↑ | NSS↑ | AUC↑ | SIM↑ | CC↑ | NSS↑ |
| UNISAL (Droste et al., 2020) | 0.901 | 0.390 | 0.490 | 2.776 | 0.934 | 0.542 | 0.673 | 3.380 | 0.917 | 0.498 | 0.636 | 3.189 |
| VSSM (Lu et al., 2023) | 0.915 | 0.383 | 0.521 | 3.027 | 0.939 | 0.583 | 0.729 | 3.927 | 0.936 | 0.560 | 0.705 | 3.908 |
| MSFF-Net (Zhang et al., 2023) | 0.913 | 0.392 | 0.534 | 3.066 | 0.940 | 0.574 | 0.723 | 3.952 | 0.933 | 0.557 | 0.698 | 3.769 |
| TFS-Net (Li et al., 2025b) | 0.912 | 0.412 | 0.527 | 2.953 | 0.934 | 0.580 | 0.725 | 3.953 | 0.930 | 0.558 | 0.664 | 3.653 |
| AAM (Ours) | 0.919 | 0.421 | 0.563 | 3.272 | 0.944 | 0.599 | 0.742 | 4.055 | 0.943 | 0.584 | 0.736 | 3.892 |

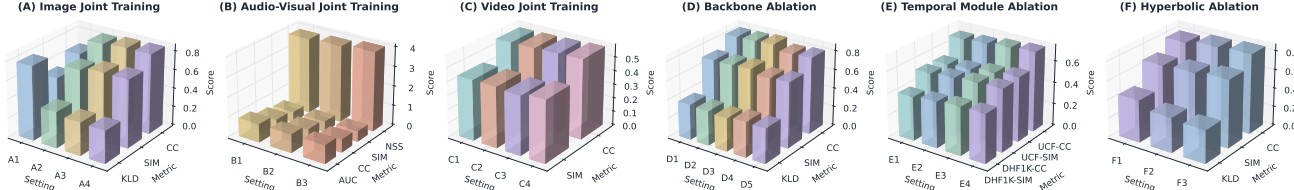

*Figure 5.* Ablation experiments: (A) Different image training settings, (B) Different audio-visual training settings, (C) Different video training settings, (D) Different backbone settings, (E) Temporal module variants, (F) Hyperbolic component ablations.

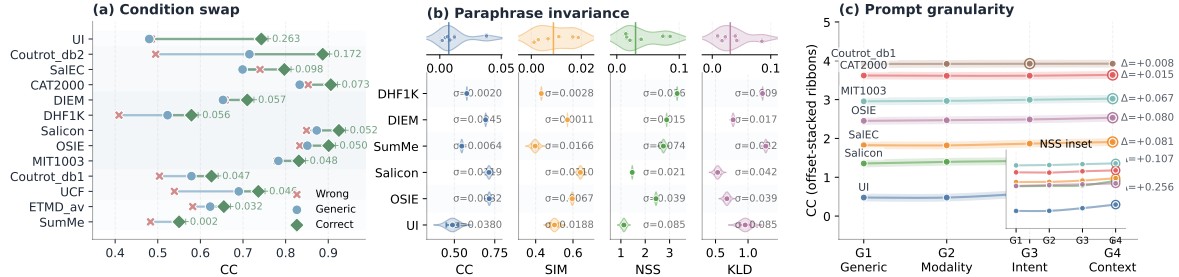

*Figure 6.* Inductive analysis of conditional attention modulation: (a) Condition swap, (b) Paraphrase invariance, (c) Prompt granularity.

*Table 4.* Comparison of complexity and efficiency metrics including Backbone, Input Length (**Fixed/Arbi**trary), Inference Speed (FPS), and **Trainable Parameters**.

| Method | Backbone | Input | FPS (img/s) | Param (M) |
|---|---|---|---|---|
| TASED (Min & Corso, 2019) | 3D Conv | Fixed | 17 | 82 |
| STSANet (Wang et al., 2021) | Video Swin | Fixed | 28 | 643 |
| TMFI-Net (Zhou et al., 2023) | Video Swin | Fixed | 30 | 234 |
| AAM (Ours) | DINOv3 | Arbi | 111 | 21.4 |

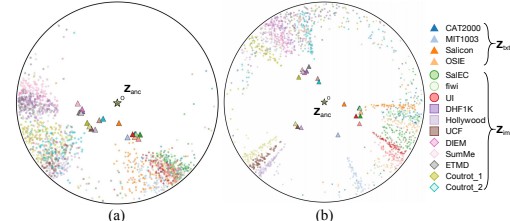

*Figure 7.* The learned hyperbolic representations: (a) HoroPCA (Chami et al., 2021) and (b) CO-SNE (Guo et al., 2022) visualizations of the latent space in $\mathbb{L}^2$.

strates that AAM treats attention as a dynamic cognitive process rather than a static, stimulus-driven pixel mapping. ❷ **Paraphrase invariance.** To verify that AAM responds to semantic content rather than specific lexical cues, we generated multiple paraphrases for each prompt, varying in wording and syntactic structure. The model exhibits invariance to surface-level linguistic variations, evidenced by the negligible performance variance (CC standard deviation $< 0.01$) across paraphrases. Such robustness indicates that AAM captures the underlying semantics of cognitive cues instead of being overfitted to specific phrasing. ❸ **Prompt granularity.** We refine prompts from general (G1) to highly specific (G4): (general→modality→viewing intention→detailed context) to probe hierarchical attention. As shown in Fig. 6 (c), performance on various tasks im-

proves with granularity, whereas dynamic and task-driven datasets plateau early. This diminishing marginal utility mirrors cognitive neuroscience findings, where dominant task contexts often override fine-grained linguistic nuances.

**Zero-Shot Generalization.** We evaluate the zero-shot generalization of our AAM on AVAD, which is unseen during training, and on LEDOV, which is trained **without text prompts**. On AVAD, AAM achieves strong performance when provided with appropriate text prompts, indicating high transferability of the learned attention representations. For LEDOV, our method also yields substantial performance gains without prompt input (Appendix 17 for details).

**Geometric Transfer Explanation.** Fig. 7 suggests that AAM organizes attention modulation on a shared hyperbolic manifold rather than independent task predictors. Generic prompts remain close to the origin, while increasingly specific task intents extend outward along hierarchical branches, providing a geometric explanation for strong zero-shot generalization across semantic conditions.

## 5. Conclusions

This paper argues that the persistent generalization failure in attention modeling is not a capacity issue, but a formulation issue: although human attention follows a unified cognitive mechanism, existing models fragment it into task-, scene-, and modality-specific problems. We address this mismatch by reframing attention variation as a hierarchical entailment process, modeling how general priors progressively specialize into task intent. Embedding this structure in hyperbolic space and modeling temporal evolution as continuous transport provide a unified geometric and physical interpretation of attention across space and time. Beyond performance gains, our results suggest that effective attention modeling requires explicit structure and dynamics, rather than isolated predictors. We hope this perspective encourages cognitively grounded foundation models that treat attention as a coherent process underlying visual perception.

## Acknowledgments

This research was supported by HUAWEI's Al Hundred Schools Program and was carried out using the Ascend AI technology stack. This work was also supported in part by the National Natural Science Foundation of China (NSFC) under Grant No. 62176169 and the Sichuan Science and Technology Program under Grant No. 2025ZNSFSC0469.

## Impact Statement

**Ethical Considerations.** We believe that our proposed AAM raises no ethical concerns regarding its motivation, design, implementation, or data usage. The method is designed to provide a unified foundation model for modeling human attention with cross-modal and cross-scene generalization while adhering to ethical guidelines in AI research.

**Societal Implications.** AAM introduces a unified perspective on human attention modeling. Unlike existing task-specific and scene-dependent methods, it captures hierarchical aspects of human attention in a cognitively motivated manner. This perspective facilitates application to real-world scenarios, improving inference efficiency while preserving output quality. Moreover, AAM provides theoretical insights and empirical evidence for the future development of holistic visual perception systems, and can serve as a basis for downstream tasks related to attention modeling and visual saliency.

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

# Contents

## A. Implementation Details

The computing infrastructure specifications are detailed in Table 5. The hyperparameter settings for AAM training are listed in Table 6.

*Table 5.* Computing infrastructure for experiments on Huawei Ascend platform.

| Component | Configuration |
|---|---|
| CPU | Huawei Kunpeng-920 (ARM64, 192 cores) |
| NPU | Huawei Ascend 910B3 (64GB HBM) |
| RAM | 1.5TB |
| OS | EulerOS 2.0 (SP10) |
| NPU Firmware | 23.0.6 |
| CANN Version | 6.3.2 |
| Language | Python 3.7 |
| Framework | MindSpore 2.1.0 |
| Dependencies | torch 2.7.1, torchvision 0.22.1, numpy 1.26.4 |

### A.1. Task Losses and Metrics

Following prior work (Wang & Shen, 2017; Wang et al., 2018; Li et al., 2025b), we utilize standard metrics for optimization and evaluation. Let $\mathbf{P}, \mathbf{G} \in \mathbb{R}^{H \times W}$ denote the predicted and ground-truth density maps (sum to 1), and $\mathbf{B} \in \{0, 1\}^{H \times W}$ denote the binary fixation map.

Table 6. Hyperparameter settings for AAM training.

| Hyperparameter | Value |
|---|---|
| *Training Dynamics* | |
| Input Resolution | $448 \times 448$ |
| Batch Size | 32 |
| Optimizer | AdamW |
| Base Learning Rate | $5 \times 10^{-4}$ |
| Weight Decay | $5 \times 10^{-3}$ |
| LR Scheduler | Cosine Annealing ($T_{max} = 10, \eta_{min} = 10^{-5}$) |
| Precision | BF16 (BFloat16) |
| *Model Architecture* | |
| Visual Backbone | DINOv3 ViT-L (Frozen) |
| Text Encoder | CLIP ViT-L/14 (Frozen) |
| Audio Encoder | Wav2CLIP Res-Net18(Frozen) |
| LoRA Rank ($r$) | 32 |
| LoRA Alpha ($\alpha$) | 64 |
| LoRA Dropout | 0.05 |
| LoRA Adaptation Scope | Last 24 Transformer Blocks |
| *Specialized Optimization* | |
| Router Weight Decay | $0.1 \times$ Base Weight Decay ($5 \times 10^{-4}$) |
| Gradient Clipping | None |

**Kullback-Leibler Divergence (KLD).** Minimizes the information loss between distributions:

$$\mathcal{L}_{\text{KLD}}(\mathbf{P}, \mathbf{G}) = \sum_i \mathbf{G}_i \log \left( \frac{\mathbf{G}_i}{\mathbf{P}_i + \epsilon} \right). \tag{24}$$

**Correlation Coefficient (CC).** Measures the linear correlation strength:

$$\mathcal{L}_{\text{CC}}(\mathbf{P}, \mathbf{G}) = \frac{\text{cov}(\mathbf{P}, \mathbf{G})}{\sigma_{\mathbf{P}} \cdot \sigma_{\mathbf{G}}}. \tag{25}$$

**Similarity Metric (SIM).** Quantifies the histogram intersection (overlap):

$$\mathcal{L}_{\text{SIM}}(\mathbf{P}, \mathbf{G}) = \sum_i \min(\mathbf{P}_i, \mathbf{G}_i). \tag{26}$$

**AUC-Judd (AUC-J).** Evaluates saliency as a binary classification task. It calculates the area under the ROC curve (TPR vs. FPR) by varying a threshold $\tau$ on $\mathbf{P}$ to classify fixation locations in $\mathbf{B}$.

## B. Dataset Details and Texts

To foster community research and facilitate the development of downstream tasks, we will publicly release the AAM codebase alongside Attention-1.75M. To support foundation model training, we curated Attention-1.75M, a standardized corpus unifying over 1.75M fixation instances across image, video, and audio-visual modalities, each equipped with dataset-level prompts. Specifically, this corpus encompasses a diverse spectrum of scenarios—including e-commerce, webpages, UIs, natural scenes, photography, portraits, and cinematic content—spanning both free-viewing and task-driven attention paradigms, as illustrated in Table. 7, Table. 8, and Table. 9. The corresponding text conditions were systematically annotated based on the detailed metadata and acquisition protocols of the source datasets, as illustrated in Fig. 8, Fig. 9, and Fig. 10. In addition, we provide summary statistics of Attention-1.75M, along with details on annotation quality control and consistency measures, in Tables 10 and 11.

*Table 7.* Summary of Image-based Attention Modeling Datasets

**🖼 Image Attention**

| Dataset | Publication | Domain | Images | Resolution | Task |
| --- | --- | --- | --- | --- | --- |
| SALICON (Huang et al., 2015) | CVPR[15] | Natural scenes | 15,000 | $640 \times 480$ | Free-view |
| MIT1003 (Judd et al., 2009) | ICCV[09] | Natural scenes | 1,003 | Varied | Free-view |
| CAT2000 (Borji & Itti, 2015) | arXiv[15] | Natural scenes | 2,000 | $1080 \times 1920$ | Free-view |
| OSIE (Xu et al., 2014) | Journal of Vision[14] | Natural scenes | 700 | $800 \times 600$ | Free-view |
| FIGRIM (Bylinskii et al., 2015) | Vision Research[15] | Natural scenes | 2,787 | $1366 \times 768$ | Free-view |
| U-EYE (Jiang et al., 2023) | ACM CHI[23] | Web pages | 1,583 | Varied | Task-driven |
| FiWI (Shen & Zhao, 2014) | ECCV[14] | Web pages | 149 | $1366 \times 768$ | Task-driven |
| SalECI (Jiang et al., 2022) | CVPR[22] | E-commerce | 871 | $720 \times 720$ | Task-driven |

*Table 8.* Summary of Video-based Attention Modeling Datasets

**🎞 Video Attention**

| Dataset | Publication | Domain | Videos | Resolution | Frames | Viewer | Task |
| --- | --- | --- | --- | --- | --- | --- | --- |
| DHF1K (Wang et al., 2018) | CVPR[18] | Natural scenes | 1,000 | $640 \times 360$ | 582,605 | 17 | Free-view |
| Hollywood-2 (Mathe & Sminchisescu, 2014) | TPAMI[15] | Movies | 1,707 | $720 \times 480$ | 487,207 | 19 | Task-driven |
| UCF Sports (Mathe & Sminchisescu, 2014) | TPAMI[15] | Sports | 150 | $720 \times 480$ | 9,900 | 19 | Task-driven |
| LEDOV (Jiang et al., 2018) | ECCV[18] | Natural scenes | 538 | $1280 \times 720$ | 179,336 | 32 | Free-view |

*Table 9.* Summary of Audio-Visual Attention Modeling Datasets

**🔊🎞 Audio-Video Attention**

| Dataset | Publication | Domain | Videos | Resolution | Frames | Viewers | Task |
| --- | --- | --- | --- | --- | --- | --- | --- |
| DIEM (Mital et al., 2011) | Cognit. Comput.[11] | Movies | 84 | $1280 \times 720$ | 240,452 | 50 | Free-view |
| Coutrot-1 (Coutrot & Guyader, 2014) | Jour. of Vision[12] | People | 60 | $1280 \times 720$ | 9,564 | 72 | Task-driven |
| Coutrot-2 (Coutrot & Guyader, 2015) | IJCV[14] | Natural scenes | 40 | $1280 \times 720$ | 25,223 | 40 | Task-driven |
| AVAD (Min et al., 2016) | TOMM[18] | Action events | 60 | $1920 \times 1280$ | 17,134 | 16 | Free-view |
| ETMD (Koutras et al., 2014) | SPIC[20] | Movies | 30 | $1280 \times 720$ | 109,788 | 25 | Free-view |
| SumMe (Gygli et al., 2014) | ECCV[14] | Sports | 25 | $640 \times 360$ | 52,744 | 15 | Task-driven |

| IMAGE | |
|---|---|
| **Datasets** | **Language-Based Conditional Cognitive Prompts** |
| **Salicon** | Static natural image from a large-scale object-in-context dataset, covering diverse everyday scenes and environments; free-viewing saliency data with weak to moderate center bias, designed for general-purpose saliency learning. |
| **CAT2000** | Static high-resolution image from a wide variety of categories including natural scenes and artificial patterns such as cartoons, sketches, fractals and abstract textures; free-viewing saliency with strong variability in category-dependent center bias. |
| **MIT1003** | Static natural photograph with strong photographer-style center framing; free-viewing eye-tracking strongly attracted to faces, people, readable text, animals and vehicles, exhibiting a strong center bias. |
| **OSIE** | Static everyday indoor or outdoor scene containing multiple interacting objects and rich semantic relationships; free-viewing eye-tracking with moderate center bias, commonly attracted to faces, gaze direction, text and object interactions. |
| **U-EYE** | Static user interface screenshot showing webpages or mobile apps, with menus, buttons, icons and dense small text; free-viewing eye-tracking where attention is strongly biased to the top-left, typical of browsing UI layouts and posters. |
| **fiwi** | Static webpage screenshot containing mixed visual elements such as text blocks, images, icons and faces in structured page layouts; free-viewing eye-tracking where attention is strongly guided by text regions typical of webpage browsing. |
| **SalEC** | Static e-commerce product image with packaging, brand logos, price tags and dense short text blocks; free-viewing eye-tracking dominated by text and logo-driven attention over retail and shopping items. |

*Figure 8.* Image attention modeling datasets and their corresponding text-based cognitive conditional prompts

| VIDEO | |
|---|---|
| **Datasets** | **Language-Based Conditional Cognitive Prompts** |
| **DHF1K** | Dynamic free-viewing video across diverse scenes and camera motions, containing multiple moving objects and complex backgrounds; saliency-style eye-tracking with dispersed attention and weak center bias. |
| **Hollywood** | Dynamic cinematic movie clip with actors, dialogues, shot changes and film-style camera motion; task-driven action recognition with strong center framing and typical movie cinematography. |
| **UCF** | Dynamic broadcast sports video showing athletes on courts or fields, often with uniforms, scoreboards and audience context; task-driven action recognition with strong center tracking of the main athlete and sports action |

*Figure 9.* Video attention modeling datasets and their corresponding text-based cognitive conditional prompts

| AUDIO-VISUAL | |
|---|---|
| **Datasets** | **Language-Based Conditional Cognitive Prompts** |
| **DIEM** | Audio-visual real-world and cinematic video clips including film, television, online video and everyday events; free-viewing eye-tracking capturing how gaze behavior shapes visual perception, memory and emotional experience over time. |
| **AVAD** | Audio-visual video clips centered on moving sound-generating objects, such as speaking faces, ; free-viewing eye-tracking emphasizing how tightly coupled audio and motion cues jointly guide visual attention over time. |
| **Coutrot_db1** | Audio-visual conversational video clips featuring multiple interacting people in realistic social environments; free-viewing eye-tracking revealing attention toward talking faces over time. |
| **Coutrot_db2** | Audio-visual video clips spanning landscapes, moving objects, and social conversations; free-viewing eye-tracking capturing how auditory cues modulate attention toward sound sources and talking faces. |
| **ETMD_av** | Audio-visual video cinematic clips from Oscar-winning films featuring high-motion action, dialogues, and complex visual storytelling; free-viewing eye-tracking showing attention guided by semantic cues such as faces over long durations. |
| **SumMe** | Audio-visual video clips depicting events like sports and holidays with static, moving, and egocentric camera motions; task-driven video summarization capturing human consensus on interestingness via explicit selection of important segments. |

*Figure 10.* Audio-visual attention modeling datasets and their corresponding text-based cognitive conditional prompts

*Table 10.* Analysis of Dataset: Summary statistics of Attention-1.75M. More detailed quantitative analysis is presented in Appendix B.

| Aspect | Quantitative Bias / Key Findings |
|---|---|
| Modality | **Static images** dominate (88.0%), while video (11.3%) and audio-visual (0.7%) data are scarce. |
| Image Scene | **Natural scenes** account for 88.3% ($>$ 70% landscape-style), while UI/web/e-commerce data contribute 11.7%. |
| Video Scene | **Professional media** (49.6%) and daily videos (43.0%) dominate, with limited sports (4.1%) and meetings (3.3%). |
| Demographic | Aged 18–36 ($\approx$ 65–70% in 18–25; $<$ 1% above 36; $>$ **50%** annotations rely on mouse-tracking proxies. |

*Table 11.* Taxonomy of Attention Levels, corresponding descriptions, and representative datasets. Building upon this hierarchy, we construct prompts using a unified standardized template.

| 📶 Level | 👁 Attention | ℹ Description | 🎛 Template Components | 🗄 Datasets |
|---|---|---|---|---|
| **High** | ⬇ Top-down | Guided by explicit goals and task objectives. | `[task setting]` + `[bias]` | Hollywood, UCF, SumMe |
| **Mid** | 📍 Semantic Mod. | Influenced by objects, interactions and semantic relationships. | `[key elements]` | OSIE, FIWI, UI, SalEC, DIEM, AVAD, Coutrot_db, ETMD_av |
| **Low** | ⬆ Bottom-up | Driven by visual stimuli such as contrast and scene statistics. | `[stimulus type]` + `[domain]` | MIT1003, CAT2000, SALICON, DHF1K |
| > _ **Standardized Prompt Template:** | | | | |
| `[stimulus type]` + `[scene/domain]` + `[key elements]` + `[task setting]` + `[attention bias]` | | | | |

# C. Data Provenance and Licensing

We carefully reviewed the terms of use for all datasets included in Attention-1.75M. To ensure transparency and responsible use, we provide a data provenance and licensing summary in this appendix, documenting the original license, usage restrictions, and redistribution permissions associated with each dataset.

**Datasets under CC BY 4.0.** The following datasets are released under the Creative Commons Attribution 4.0 International license (CC BY 4.0): SALICON, U-EYE, DHF1K, Hollywood-2, and UCF. These datasets allow reuse and redistribution with appropriate attribution, subject to the terms of the license.

**Research-only datasets.** Several datasets are made available for research purposes only, following the terms specified in their original publications or official release pages. These include MIT300, CAT2000, FIGRIM, FiWI, LEDOV, Coutrot_db, DIEM, ETMD, and SumMe. For these datasets, we follow the original usage restrictions and do not claim any additional redistribution rights.

**Datasets under the MIT License.** OSIE and SalECI are released under the MIT License, which permits reuse, modification, and redistribution under the conditions specified by the license.

**Aggregate benchmark policy.** Attention-1.75M is constructed as an aggregate benchmark from datasets with heterogeneous licensing conditions. To ensure compliance with the most restrictive components, Attention-1.75M will be released under a research-only policy.

**Licensing matrix.** Table 12 summarizes the licensing status and redistribution policy for each dataset.

# D. Theoretical Proof and Analysis

### D.1. Theoretical Properties of the Discretized FP Dynamics

The Fokker–Planck equation provides a physically grounded framework for modeling temporal saliency evolution, while enforcing a key structural constraint via **mass conservation**. This constraint stabilizes training and promotes meaningful, interpretable attention dynamics.

*Table 12.* Data provenance and licensing matrix for the datasets included in Attention-1.75M.

| Dataset | License / Usage Policy | Our Redistribution Policy |
|---|---|---|
| SALICON | CC BY 4.0 | Refer to original license terms |
| U-EYE | CC BY 4.0 | Refer to original license terms |
| DHF1K | CC BY 4.0 | Refer to original license terms |
| Hollywood-2 | CC BY 4.0 | Refer to original license terms |
| UCF | CC BY 4.0 | Refer to original license terms |
| MIT300 | Research only | Not redistributed |
| CAT2000 | Research only | Not redistributed |
| FIGRIM | Research only | Not redistributed |
| FiWI | Research only | Not redistributed |
| LEDOV | Research only | Not redistributed |
| Coutrot_db | Research only | Not redistributed |
| DIEM | Research only | Not redistributed |
| ETMD | Research only | Not redistributed |
| SumMe | Research only | Not redistributed |
| OSIE | MIT License | Refer to original license terms |
| SalECI | MIT License | Refer to original license terms |

In video saliency prediction, standard metrics (e.g., KL, CC, SIM) implicitly assume attention to be a conserved finite resource. Accordingly, we model the saliency state at time $t$ as a probability density on $\Omega$ and enforce the simplex constraint:

$$u_t \in \Delta = \left\{ u \succeq 0 \;\middle|\; \sum_{x \in \Omega} u(x) = 1 \right\}. \tag{27}$$

Without this conservation law, the temporal module could trivially reduce loss by globally rescaling saliency magnitude, rather than capturing motion-driven drift and diffusion. Mass conservation removes this degree of freedom, ensuring stable and physically meaningful evolution.

In this appendix, we formally analyze the theoretical properties of the resulting discrete system. Specifically, we show that the Lie–Trotter operator splitting scheme (Sec. 3.4) preserves the probabilistic interpretation of attention distributions and ensures numerical stability.

**Probability simplex.** Recall that each saliency state $u_t \in \mathbb{R}^{|\Omega|}$ is defined on the probability simplex $\Delta := \left\{ u \succeq 0, \; \mathbf{1}^\top u = 1 \right\}$.

**Proposition D.1** (Simplex Invariance and Stability). *Assume periodic or zero-flux boundary conditions along the temporal axis. If the initial state satisfies $u_t \in \Delta$ and the diffusion step size obeys the CFL condition*

$$\Delta t\, \nu_t(x) \le \frac{1}{2}, \qquad \forall\, t, x, \tag{28}$$

*then the discretized FP evolution satisfies:*

1. **Non-negativity:** *all intermediate states remain element-wise non-negative.*

2. **Mass boundedness:** *the $\ell_1$ mass remains bounded and is strictly normalized after projection.*

3. **Simplex invariance:** *the projection operator guarantees $u_t \in \Delta$ at every iteration.*

*Proof Sketch. Drift step.* The temporal attention kernel $A_x(t \leftarrow t')$ is row-stochastic, thus defining a valid Markov transition. Consequently, the drift update is a convex combination of non-negative states and preserves non-negativity.

*Diffusion step.* The explicit second-order diffusion update can be rewritten as a convex interpolation of neighboring temporal states. Under the CFL constraint $\Delta t \nu_t \le 1/2$, all coefficients remain non-negative, ensuring stability and preventing oscillatory divergence.

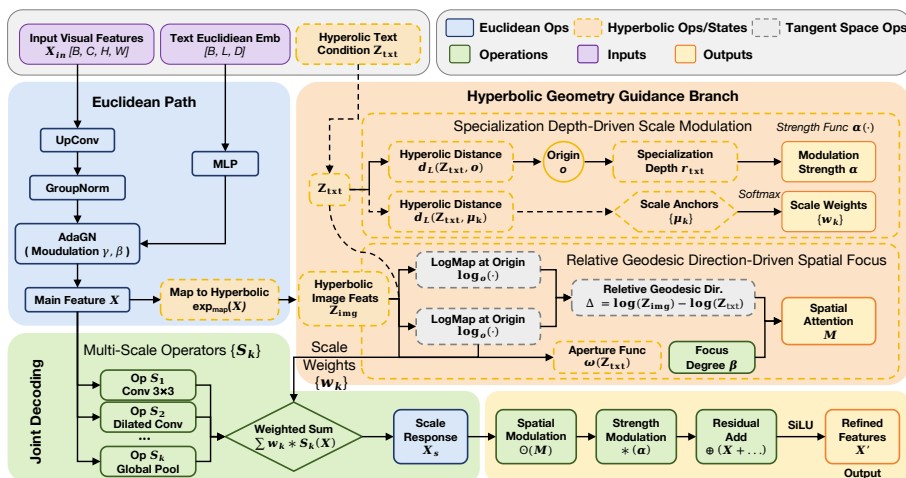

*Figure 11.* The architecture of audio fusion module

*Correction step.* The correction operator interpolates between the predicted state and the observed distribution, hence preserving non-negativity.

*Projection.* Finally, the simplex projection explicitly enforces $\mathbf{1}^\top u_t = 1$, eliminating accumulated numerical drift and ensuring a valid probability distribution for the next iteration. $\square$

### D.2. Supplementary Derivation Details for FPD Parameters

The drift equation is given by $\frac{\partial u}{\partial t} = -\nabla_\tau \cdot (\mathbf{v}u)$. To simulate this physical process within a discrete feature space, we utilize bidirectional temporal self-attention to parameterize the drift propagator $A_x$. We define the discrete transition kernel $A_x(t \leftarrow t')$ from source time $t'$ to target time $t$ as:

$$A_x^{(h)}(t \leftarrow t') = \frac{\exp\left(\langle q_t^{(h)}(x), k_{t'}^{(h)}(x)\rangle/(\sqrt{d}\beta)\right)}{\sum_{\tau'=1}^{T} \exp\left(\langle q_t^{(h)}(x), k_{\tau'}^{(h)}(x)\rangle/(\sqrt{d}\beta)\right)}. \tag{29}$$

Here, $x \in \Omega$ denotes a fixed spatial location, $t$ represents the target time step for the current update, and $t'$ indicates the source time step providing information. $q_t^{(h)}$ and $k_{t'}^{(h)}$ refer to the query and key projections of the $h$-th attention head, respectively. The summation term in the denominator functions as a normalization factor, where $\tau'$ is a dummy index traversing the entire temporal axis $1, \ldots, T$. Finally, $\beta$ serves as a temperature coefficient to modulate the entropy of the transition distribution.

## E. Methodology Details

### E.1. Hyperbolic Decoder

As shown in Fig. 11 and Algorithm 1, the decoder modulates the backbone feature $X \in \mathbb{R}^{B \times C \times H \times W}$ using the text tangent vector $t$ and hyperbolic condition embedding $z_{\text{cond}}$. First, we compute the condition depth $r_{\text{cond}} = d_{\mathbb{L}}(z_{\text{cond}}, 0)$ to control the modulation strength. This scalar guides the **pixel-level hyperbolic gating**, where we apply a log–gate–exp transformation: $\tilde{X} = \text{HypPixGate}(X, r_{\text{cond}})$. Next, for **multi-scale extraction**, we obtain structural responses $\{Y_k\}_{k=1}^K = \{\mathcal{O}_k(\tilde{X})\}$ using local, dilated, and global operators. These scales are fused via an anchor-based module:

$$y = \text{HypScaleFuse}(\{Y_k\}, t, z_{\text{cond}}), \tag{30}$$

where weights depend on text features and hyperbolic distances. Finally, the fused term $y$ is injected into the **upsampling blocks** as $\hat{X} = \text{UpBlock}(X, t, y, r_{\text{cond}})$, and the final saliency map is generated by a **joint Gaussian bias head**:

$$S = \text{GaussianBiasHead}(\hat{X}, t, z_{\text{cond}}, r_{\text{cond}}). \tag{31}$$

---

**Algorithm 1** Hyperbolic Multi-Scale Decoder (Data Flow)

---

**Input:** backbone feature map $X$
**Input:** text feature $t$
**Input:** hyperbolic condition point $z_{\mathrm{cond}}$
**Output:** saliency map $S$

**Step 1: Condition depth**
$r_{\mathrm{cond}} \leftarrow d_{\mathbb{L}}(z_{\mathrm{cond}}, 0)$

**Step 2: Pixel-level hyperbolic modulation**
$\tilde{X} \leftarrow \mathrm{HypPixGate}(X, r_{\mathrm{cond}})$ {log–gate–exp conditional gating}

**Step 3: Multi-scale structural extraction**
$\{Y_k\}_{k=1}^K \leftarrow \mathrm{MultiScaleOps}(\tilde{X})$ {local / dilated / global operators}

**Step 4: Hyperbolic scale fusion**
$y \leftarrow \mathrm{HypScaleFuse}(\{Y_k\}, t, z_{\mathrm{cond}})$ {anchor-distance scale gating}

**Step 5: Upsampling with depth-controlled injection**
$\hat{X} \leftarrow \mathrm{UpBlock}(X, t, y, r_{\mathrm{cond}})$ {Ada-style modulation + residual injection}

**Step 6: Joint Gaussian bias prediction**
$S \leftarrow \mathrm{GaussianBiasHead}(\hat{X}, t, z_{\mathrm{cond}}, r_{\mathrm{cond}})$

**Return:** $S$

---

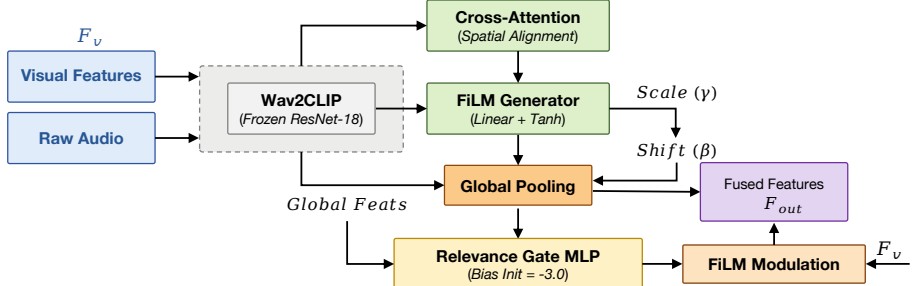

*Figure 12.* The architecture of audio-visual fusion branch

### E.2. Audio-Visual Branch with Robust Feature Modulation

To effectively exploit auditory semantics while mitigating the interference from irrelevant background noise (i.e., the audio-visual mismatch problem), we introduce a lightweight yet robust audio-visual fusion branch, as illustrated in Fig. 12 and Algorithm 2.

**Audio Representation.** To ensure semantic alignment with the visual modality under a limited computational budget, we avoid using computationally expensive Transformer-based audio encoders. Instead, we adopt **Wav2CLIP** (Wu et al., 2022) as our audio feature extractor. Wav2CLIP employs a ResNet-18 backbone distilled from the CLIP image encoder, projecting the raw audio waveform $\mathcal{A} \in \mathbb{R}^{T \times L}$ into a semantic embedding sequence $\mathbf{F}_a \in \mathbb{R}^{T \times C_a}$, where $C_a = 512$. This design naturally places the extracted audio features in a shared latent space with the visual representations, facilitating cross-modal interaction.

**Spatial Alignment via Cross-Attention.** Given the visual feature map from the backbone network, denoted as $\mathbf{F}_v \in \mathbb{R}^{T \times HW \times C_v}$, we first establish spatial correspondence across modalities. Specifically, we employ multi-head cross-attention (MHCA), where the flattened visual features $\mathbf{F}_v$ serve as the *Query*, while the audio embeddings $\mathbf{F}_a$ act as both the *Key* and *Value*. This operation produces a spatially aligned auditory representation $\hat{\mathbf{F}}_a$, which guides auditory context toward visually relevant regions.

**Stabilized FiLM Fusion.** Naïve additive fusion often leads to the phenomenon of *modality laziness*, where one modality dominates while the other is ignored. Instead, we adopt **Feature-wise Linear Modulation (FiLM)** to conditionally modulate the visual statistics using audio cues. To improve training stability, we propose a *Residual Tanh-FiLM* mechanism.

---

**Algorithm 2** Robust Audio-Visual Fusion Branch with Stabilized Modulation

---

**Input:** audio waveform $\mathcal{A} \in \mathbb{R}^{T \times L}$
**Input:** visual features $\mathbf{F}_v \in \mathbb{R}^{T \times HW \times C_v}$
**Output:** fused visual features $\mathbf{F}_{\text{out}}$

**Step 1: Audio representation (Wav2CLIP)**
$\mathbf{F}_a \leftarrow \text{Wav2CLIP}(\mathcal{A})$ {$\mathbf{F}_a \in \mathbb{R}^{T \times C_a}$, aligned with CLIP space}

**Step 2: Spatial alignment via cross-attention**
$\hat{\mathbf{F}}_a \leftarrow \text{MHCA}(\mathbf{Q} = \mathbf{F}_v, \ \mathbf{K} = \mathbf{F}_a, \ \mathbf{V} = \mathbf{F}_a)$ {audio context aligned to visual spatial locations}

**Step 3: Modality gating (audio-visual relevance)**
$\alpha \leftarrow \sigma(\text{MLP}_{\text{gate}}([\text{Pool}(\mathbf{F}_v) \| \text{Pool}(\mathbf{F}_a)]))$ {$\alpha \in [0, 1]$, initialized with visual-prior bias}

**Step 4: Stabilized Residual Tanh-FiLM fusion**
$(\gamma, \beta) \leftarrow \text{Linear}(\hat{\mathbf{F}}_a)$ {zero-initialized projection}
$\mathbf{F}_{\text{out}} \leftarrow \mathbf{F}_v \odot \left(1 + \alpha \cdot \tanh(\gamma)\right) + \alpha \cdot \beta$ {bounded modulation prevents gradient explosion}

**Step 5: Negative Sample Attack (training only)**
**if** with probability $p = 0.3$ **then**
    Replace $\mathcal{A}$ with Gaussian noise
**end if**
Add sparsity regularization loss: $\mathcal{L}_{\text{gate}} = \|\alpha\|_2$
**Return:** $\mathbf{F}_{\text{out}}$

---

Concretely, we generate the scaling and shifting parameters $(\gamma, \beta)$ from $\hat{\mathbf{F}}_a$ through a zero-initialized linear projection layer. The modulation is formulated as:

$$\mathbf{F}_{out} = \mathbf{F}_v \odot (1 + \alpha \cdot \tanh(\gamma)) + (\alpha \cdot \beta), \tag{32}$$

where $\odot$ denotes element-wise multiplication. The $\tanh(\cdot)$ function bounds the scaling factor within a stable range, effectively preventing gradient explosion during early training.

**Modality Gating and Negative Sample Attack.** To explicitly handle audio-visual mismatch, we introduce a learnable relevance gate $\alpha \in [0, 1]$, computed by a lightweight MLP that takes global audio-visual embeddings as input. We initialize the gate bias to $-3.0$ (yielding an initial $\alpha \approx 0.05$), thereby imposing a *visual-prior* at the beginning of training. Furthermore, we adopt a **Negative Sample Attack (NSA)** strategy: with probability $p = 0.3$, the audio input is replaced by Gaussian noise, and a sparsity regularization term $\|\alpha\|_2$ is applied to encourage suppression of unreliable auditory signals. This adversarial setup forces the network to explicitly recognize and reject irrelevant or noisy audio cues.

# F. Additional Results

Synthesizing the ablation studies discussed in the main text, we draw the following conclusions:

1. Given visual input, the attention distribution is not unique but systematically modulated by viewing conditions and cognitive contexts.

2. Language prompts provide an effective and interpretable mechanism for characterizing this modulation.

3. Through hierarchical and unified modeling, shared structures across different modalities and attention domains are effectively captured.

4. Collectively, these designs significantly enhance the model's generalization capabilities across datasets and zero-shot scenarios.

In this appendix, we provide more detailed quantitative results and analyses.

## F.1. Detailed Ablation Studies

**Joint Training Efficacy.** Tables 13 and 14 present detailed results for image and audio-visual joint training, respectively. The results on image datasets demonstrate that, under our hierarchical modeling framework, joint multi-source attention training further boosts performance. Conversely, unconditional modeling leads to a significant performance degradation.

**Backbone and Modality Analysis.** Table 15 provides a quantitative comparison of different visual backbones, while Table 16 details the ablation results for the audio component, confirming that the incorporation of audio cues effectively aids attention prediction in multimodal scenarios.

## F.2. Inductive Analysis

To verify whether AAM effectively captures the hierarchical behaviors of cognitive modulation, we analyze the mechanism from three complementary perspectives and validate the model's generalization ability (see Table 18 for comprehensive results): ❶ Condition Swap, ❷ Paraphrase Invariance, ❸ Prompt Granularity.

❶ **Condition Swap: Attention Is Not Input-Only.** For each dataset, we compare three condition settings: i) **Correct condition** (prompts aligned with the dataset protocol); ii) **No condition** (generic free-viewing prompts); and iii) **Wrong condition** (mismatched tasks).

❷ **Paraphrase Invariance: Conditioning on Semantics.** To verify that the linguistic conditioning operates at a semantic level rather than relying on fixed lexical cues, we generated multiple paraphrases for each prompt. These paraphrases vary in wording, length, and syntactic structure but preserve the underlying semantic meaning (see Fig. 13).

❸ **Prompt Granularity: Hierarchical Cognitive Conditioning.** We further investigate whether attention modulation exhibits a hierarchical structure by progressively refining prompts from general to highly specific. For each dataset, we constructed a four-level prompt hierarchy (see Fig. 14):

- **G1 (General):** Free-viewing without task or context specification.

- **G2 (Modality):** Specifying image, video, or audio-visual input.

- **G3 (Viewing Intention):** Coarse alignment with the dataset's viewing protocol.

- **G4 (Detailed Context):** Full scene description and dominant attention drivers.

Across extensive benchmarks, we observe a consistent trend: semantics- and layout-driven datasets (e.g., Salicon, OSIE, U-EYE, MIT1003, SalEC) show significant performance gains as prompt granularity increases. Dynamic and audio-visual datasets (e.g., DHF1K, DIEM) exhibit modest but stable gains, indicating that linguistic modulation is non-negligible. Task-driven datasets (e.g., UCF, SumMe) display dataset-specific optimal granularities, reflecting the dominance of task structure in guiding attention. Performance improves when prompt granularity matches the true viewing protocol and cognitive context, whereas over-specification or mis-specification yields diminishing returns. This behavior aligns with findings in cognitive neuroscience, suggesting that attention is jointly shaped by general viewing priors and specific task goals.

## F.3. Zero-shot Generalization

As shown in Table 17, we evaluate the zero-shot generalization of AAM on AVAD (unseen during training) and LEDOV (trained without text inputs). On AVAD, AAM matches the performance of state-of-the-art task-specific models under both text-conditioned and unconditional settings. On the diverse and large-scale LEDOV benchmark, AAM achieves significant performance gains without text input (see Table 21), demonstrating robust real-world applicability and generalization ability.

### F.3.1. VISUALIZATION OF HYPERBOLIC SPACE CHARACTERISTICS

**Fig. 7 visualizes the learned hyperbolic embeddings, revealing a distinct "general → context → instance" radial hierarchy.** Driven by the hierarchical ordering constraints, the unconditional anchors stably cluster near the origin, while conditional text embeddings (▲) and specific instance embeddings (●/■/♦) are pushed progressively toward the boundary.

*Figure 13.* Inductive analysis of conditional attention modulation: (b) Paraphrase invariance.

This radial stratification utilizes the exponential volume growth of hyperbolic space to prevent collapse and accommodate the vast diversity of specific samples. **Crucially, the angular distribution exhibits an "intertwined yet separable" structure governed by multi-factor interactions:** while modalities (Image/Video/Audio-Visual) form high-level directional branches, distinct viewing intentions (free-viewing vs. task-driven) induce significant intra-modal separation, and shared attention cues (e.g., faces or motion) create local cross-modal proximity without merging. **This structure is quantitatively corroborated by the norm distribution analysis (Fig. 7),** which confirms that our hyperbolic entailment cones successfully encode attention modeling as a coherent geometric process capturing radial specialization, angular semantic containment, and hierarchical differentiation.

## F.4. More Comparison Results

Due to space constraints, we present only a subset of state-of-the-art (SOTA) methods in the main text. For a more comprehensive comparison, we here evaluate our model against a broader range of baselines across Image Attention Modeling (Table 19), Video Attention Modeling (Tables 20 and 21), and Audio-Visual Modeling (Table 22). Additionally, we include qualitative visualization results under diverse scenarios and task settings in Fig. 15, Fig. 16

## G. Future Work.

**LLM-Driven Hierarchical Refinement.** Currently, textual conditions are primarily derived from fixed dataset labels or coarse captions. In future work, we plan to integrate **Large Language Models (LLMs)** to synthesize fine-grained, open-vocabulary descriptions. By leveraging the reasoning capabilities of LLMs, we aim to construct richer semantic trees, enabling the model to capture more subtle, granular hierarchical relationships in human attention modulation.

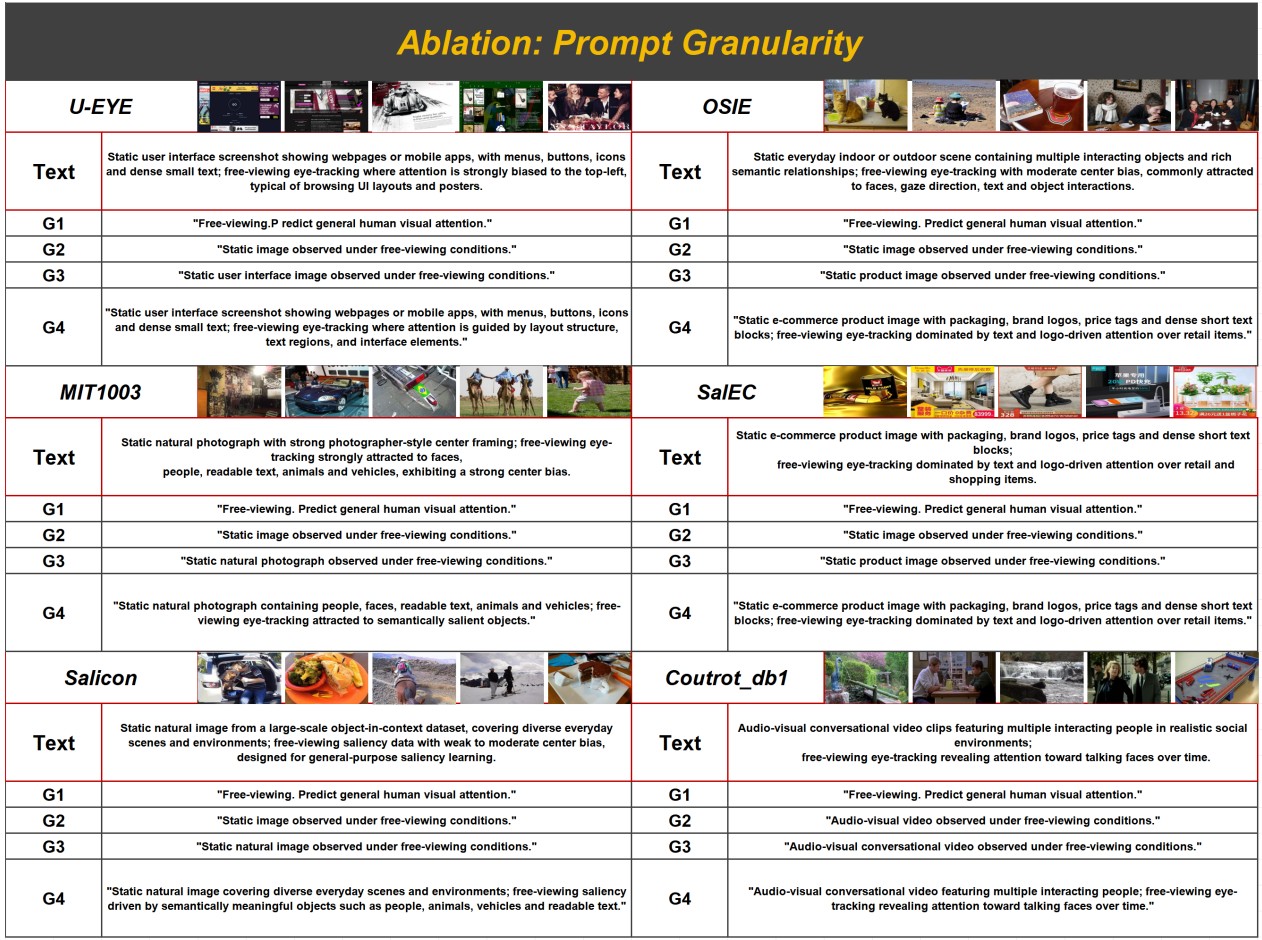

*Figure 14.* Inductive analysis of conditional attention modulation: (c) Prompt granularity.



(a) Image Attention Modeling  (b) Video Attention Modeling  (c) Audio-Visual Modeling

*Figure 15.* Visual comparison against SOTA methods.

**Universal Foundation for Downstream Tasks.**  Furthermore, we envision AAM serving as a versatile **foundation model** for the broader landscape of human attention and saliency-related research. We propose that diverse downstream tasks can be effectively unified under our paradigm, allowing for consistent modeling across different applications. Establishing AAM as a general-purpose backbone for these tasks represents a pivotal direction for our future research.

*Table 13.* Quantitative comparison of different training strategies on six image saliency datasets. Results demonstrate the effectiveness of joint training and the proposed hyperbolic components. Best results are highlighted in **bold**.

| Method | 🖼 Image Joint training | | | | | | | | | | | | | | | | | |
|---|---|---|---|---|---|---|---|---|---|---|---|---|---|---|---|---|---|---|
| | Salicon | | | OSIE | | | U-EYE | | | MIT1003 | | | SalECI | | | CAT2000 | | |
| | CC↑ | SIM↑ | KLD↓ | CC↑ | SIM↑ | KLD↓ | CC↑ | SIM↑ | KLD↓ | CC↑ | SIM↑ | KLD↓ | CC↑ | SIM↑ | KLD↓ | CC↑ | SIM↑ | KLD↓ |
| Single Training | 0.911 | 0.806 | 0.175 | 0.872 | 0.727 | 0.312 | 0.733 | 0.630 | 0.539 | 0.781 | 0.622 | 0.537 | 0.788 | 0.673 | 0.469 | 0.879 | 0.633 | 0.270 |
| Cross-Dataset Test | 0.786 | 0.625 | 0.485 | 0.743 | 0.596 | 0.567 | 0.365 | 0.424 | 1.222 | 0.696 | 0.471 | 0.901 | 0.611 | 0.461 | 0.950 | 0.723 | 0.585 | 0.617 |
| Joint Image (w/o Hyp) | 0.907 | 0.796 | 0.191 | 0.870 | 0.724 | 0.331 | 0.639 | 0.578 | 0.709 | 0.777 | 0.604 | 0.571 | 0.763 | 0.663 | 0.521 | 0.820 | 0.708 | 0.361 |
| Joint Image (w/ Hyp) | 0.917 | 0.808 | 0.179 | 0.881 | 0.740 | 0.288 | 0.741 | 0.633 | 0.531 | 0.794 | 0.643 | 0.509 | 0.790 | 0.673 | 0.462 | 0.887 | 0.679 | 0.255 |
| Joint Image (w/ Hyp + Dec) | 0.924 | 0.816 | 0.166 | 0.900 | 0.760 | 0.249 | **0.746** | **0.635** | 0.527 | 0.824 | **0.674** | 0.469 | 0.796 | 0.677 | 0.459 | **0.909** | **0.771** | **0.238** |
| Image+Video (w/o Hyp) | 0.901 | 0.789 | 0.201 | 0.866 | 0.724 | 0.337 | 0.607 | 0.522 | 0.852 | 0.779 | 0.611 | 0.572 | 0.674 | 0.455 | 0.887 | 0.806 | 0.698 | 0.363 |
| Image+Video (w/ Hyp) | **0.925** | **0.819** | **0.163** | **0.901** | **0.762** | **0.245** | 0.745 | **0.635** | **0.523** | **0.830** | **0.674** | **0.445** | **0.796** | **0.678** | **0.452** | 0.907 | 0.769 | 0.241 |

*Table 14.* Ablation study of audio-video joint training across five audio-visual saliency datasets. We progressively incorporate additional video and image data into the training process. Results show that multi-source joint training consistently improves performance. Best results are highlighted in **bold**.

| Method | 🔊⊞ Audio-Video Joint training | | | | | | | | | | | | | | | | | | | |
|---|---|---|---|---|---|---|---|---|---|---|---|---|---|---|---|---|---|---|---|---|
| | DIEM | | | | Coutrot_db1 | | | | Coutrot_db2 | | | | ETMD_av | | | | SumMe | | | |
| | AUC↑ | CC↑ | SIM↑ | NSS↑ | AUC↑ | CC↑ | SIM↑ | NSS↑ | AUC↑ | CC↑ | SIM↑ | NSS↑ | AUC↑ | CC↑ | SIM↑ | NSS↑ | AUC↑ | CC↑ | SIM↑ | NSS↑ |
| AV | 0.910 | 0.662 | 0.535 | 2.69 | 0.902 | 0.589 | 0.464 | 2.95 | 0.970 | 0.886 | 0.698 | 7.360 | 0.939 | 0.531 | 0.482 | 3.485 | 0.910 | 0.521 | 0.393 | 2.744 |
| AV + Video | 0.916 | 0.684 | 0.489 | 2.51 | 0.906 | 0.625 | 0.489 | 3.15 | 0.971 | 0.888 | 0.706 | 7.410 | 0.941 | 0.546 | 0.500 | 3.596 | 0.915 | 0.535 | 0.406 | 2.832 |
| AV + Video + Image | **0.919** | **0.710** | **0.572** | **2.88** | **0.911** | **0.626** | **0.496** | **3.22** | **0.971** | **0.887** | **0.697** | **7.46** | **0.945** | **0.550** | **0.504** | **3.66** | **0.920** | **0.550** | **0.420** | **2.90** |

*Table 15.* Ablation study of different backbones. The best results are shown in **bold**.

| Model | ⌨ Ablation Study of Foundation Models | | | | | | | | | | | | | | | | | |
|---|---|---|---|---|---|---|---|---|---|---|---|---|---|---|---|---|---|---|
| | Salicon | | | OSIE | | | U-EYE | | | MIT1003 | | | SalECI | | | CAT2000 | | |
| | CC↑ | SIM↑ | KLD↓ | CC↑ | SIM↑ | KLD↓ | CC↑ | SIM↑ | KLD↓ | CC↑ | SIM↑ | KLD↓ | CC↑ | SIM↑ | KLD↓ | CC↑ | SIM↑ | KLD↓ |
| DINOv3_small | 0.918 | 0.811 | 0.177 | 0.857 | 0.727 | 0.317 | 0.726 | 0.624 | 0.550 | 0.800 | 0.650 | 0.508 | **0.800** | **0.686** | 0.447 | 0.894 | 0.760 | 0.256 |
| DINOv3_base | 0.920 | 0.810 | 0.172 | 0.869 | 0.734 | 0.303 | 0.729 | 0.624 | 0.550 | 0.811 | 0.656 | 0.477 | 0.795 | 0.678 | 0.443 | 0.897 | 0.762 | 0.247 |
| DINOv3_large | **0.924** | **0.816** | **0.166** | **0.900** | **0.760** | **0.249** | **0.746** | **0.635** | **0.527** | **0.824** | **0.674** | **0.469** | 0.796 | 0.677 | 0.459 | **0.909** | **0.771** | **0.238** |
| SAM2_base | 0.871 | 0.763 | 0.243 | 0.811 | 0.722 | 0.289 | 0.719 | 0.613 | 0.566 | 0.790 | 0.641 | 0.500 | 0.729 | 0.613 | **0.397** | 0.870 | 0.726 | 0.245 |
| SAM2_large | 0.884 | 0.788 | 0.211 | 0.824 | 0.739 | 0.266 | 0.731 | 0.624 | 0.545 | 0.798 | 0.644 | 0.515 | 0.742 | 0.625 | 0.404 | 0.881 | 0.739 | 0.266 |

*Table 16.* Ablation study of the audio modality.

| Setting | 🔊 Impact of Audio Modality | | | | | | | | | | | | | | |
|---|---|---|---|---|---|---|---|---|---|---|---|---|---|---|---|
| | DIEM | | | Coutrot_db1 | | | Coutrot_db2 | | | ETMD_av | | | SumMe | | |
| | CC↑ | SIM↑ | NSS↑ | CC↑ | SIM↑ | NSS↑ | CC↑ | SIM↑ | NSS↑ | CC↑ | SIM↑ | NSS↑ | CC↑ | SIM↑ | NSS↑ |
| w/o Audio | 0.703 | 0.571 | **2.880** | 0.615 | 0.492 | 3.180 | 0.862 | 0.689 | 7.40 | 0.648 | 0.422 | **3.660** | 0.549 | **0.422** | **2.910** |
| w/ Audio | **0.710** | **0.572** | **2.880** | **0.626** | **0.496** | **3.223** | **0.887** | **0.697** | **7.46** | **0.655** | **0.504** | 3.656 | **0.550** | 0.420 | 2.895 |

*Table 17.* Performance on Zero-shot Generalization.

| 🌐 Zero-shot Generalization | | | | |
|---|---|---|---|---|
| **Method** | **AUC-J↑** | **NSS↑** | **CC↑** | **SIM↑** |
| **AVAD Dataset** | | | | |
| MSPI (Xie et al., 2024) | 0.935 | 3.870 | 0.697 | 0.529 |
| TAVDiff (Yu et al., 2025) | 0.949 | 4.290 | 0.729 | 0.550 |
| AAM (Ours) (w/ Text) | 0.936 | 4.101 | 0.712 | 0.557 |
| AAM (Ours) (Zero-shot, w/o Text) | 0.933 | 4.100 | 0.695 | 0.542 |
| **Method** | **AUC-J↑** | **NSS↑** | **CC↑** | **KLD↑** |
| **LEDOV Dataset** | | | | |
| Sal-DCNN (Jiang et al., 2019) | 0.892 | 2.838 | 0.573 | 1.304 |
| AAM (Ours)(w/o Text) | 0.941 | 3.652 | 0.708 | 0.888 |

*Table 18.* Prompt-related ablations and robustness analyses. (a) Condition Swap / Text Prompt Impact. (b) Paraphrase Invariance. (c) Prompt Granularity. Best results are in **bold**.

| (a) ⌨ Condition Swap / Text Prompt Impact | | | | | | | | | | | | | | | | | | | | |
|---|---|---|---|---|---|---|---|---|---|---|---|---|---|---|---|---|---|---|---|---|
| **Setting** | **Salicon** | | | **OSIE** | | | **UI** | | | **MIT1003** | | | **SalECI** | | | **CAT2000** | | | **DHF1K** | | |
| | CC↑ | SIM↑ | KLD↓ | CC↑ | SIM↑ | KLD↓ | CC↑ | SIM↑ | KLD↓ | CC↑ | SIM↑ | KLD↓ | CC↑ | SIM↑ | KLD↓ | CC↑ | SIM↑ | KLD↓ | CC↑ | SIM↑ | NSS↑ |
| Correct Text Input | **0.925** | **0.819** | **0.163** | **0.901** | **0.760** | **0.243** | **0.743** | **0.635** | **0.524** | **0.831** | **0.674** | **0.446** | **0.797** | **0.678** | **0.450** | **0.906** | **0.769** | **0.235** | **0.579** | **0.421** | **3.272** |
| No Text Input (General) | 0.873 | 0.759 | 0.247 | 0.851 | 0.703 | 0.375 | 0.480 | 0.494 | 1.060 | 0.783 | 0.620 | 0.549 | 0.699 | 0.589 | 0.618 | 0.833 | 0.719 | 0.361 | 0.523 | 0.368 | 2.960 |
| Wrong Prompt (Cross-Task) | 0.849 | 0.736 | 0.278 | 0.833 | 0.686 | 0.389 | 0.485 | 0.498 | 0.968 | 0.780 | 0.614 | 0.551 | 0.740 | 0.612 | 0.557 | 0.853 | 0.736 | 0.308 | 0.409 | 0.288 | 2.306 |

| (b) ✏ Paraphrase Invariance | | | | | | |
|---|---|---|---|---|---|---|
| **Metric** | **DHF1K** | **DIEM** | **SumMe** | **Salicon** | **OSIE** | **UI** |
| CC | $0.5768 \pm 0.0020$ | $0.6933 \pm 0.0045$ | $0.5458 \pm 0.0064$ | $0.7141 \pm 0.0119$ | $0.7158 \pm 0.0082$ | $0.4883 \pm 0.0380$ |
| SIM | $0.4302 \pm 0.0028$ | $0.5702 \pm 0.0011$ | $0.3971 \pm 0.0166$ | $0.6395 \pm 0.0110$ | $0.5961 \pm 0.0067$ | $0.5008 \pm 0.0188$ |
| NSS | $3.3121 \pm 0.0158$ | $2.8782 \pm 0.0145$ | $2.7465 \pm 0.0738$ | $1.4653 \pm 0.0211$ | $2.4431 \pm 0.0390$ | $1.1225 \pm 0.0854$ |
| KLD | $1.2227 \pm 0.0092$ | $0.7708 \pm 0.0167$ | $1.2722 \pm 0.0223$ | $0.5325 \pm 0.0420$ | $0.6748 \pm 0.0393$ | $0.9564 \pm 0.0849$ |

| (c) Prompt Granularity Ablation (G1 → G4) | | | | | | | | | | | | | | | |
|---|---|---|---|---|---|---|---|---|---|---|---|---|---|---|---|
| **CC ↑** | | | | | **NSS ↑** (Static) | | | | | **NSS ↑** (Video) | | | | |
| Dataset | G1 | G2 | G3 | G4 | Dataset | G1 | G2 | G3 | G4 | Dataset | G1 | G2 | G3 | G4 |
| Salicon | 0.805 → | 0.848 → | 0.867 → | **0.913** | Salicon | 1.722 → | 1.804 → | 1.865 → | **1.958** | DHF1K | 0.576 → | 0.577 → | 0.577 → | **0.579** |
| OSIE | 0.803 → | 0.818 → | 0.839 → | **0.883** | OSIE | 2.797 → | 2.828 → | 2.864 → | **3.291** | UCF | 0.742 → | 0.742 → | **0.743** → | 0.741 |
| UI | 0.478 → | 0.477 → | 0.586 → | **0.734** | UI | 1.097 → | 1.104 → | 1.347 → | **1.676** | DIEM | 0.700 → | 0.698 → | 0.697 → | **0.702** |
| MIT1003 | 0.756 → | 0.766 → | 0.795 → | **0.824** | MIT1003 | 2.624 → | 2.638 → | 2.762 → | **2.974** | SumMe | 0.547 → | 0.550 → | 0.549 → | **0.552** |
| SalECI | 0.726 → | 0.721 → | 0.767 → | **0.807** | SalECI | 1.861 → | 1.833 → | 1.954 → | **2.038** | CAT2000 | 0.871 → | 0.863 → | 0.866 → | **0.886** |

*Table 19.* Quantitative comparison on **image** attention modeling datasets. The best results are shown in red, and the second best in blue.

| 🖼 Image Attention Modeling | | | | | | | | | |
|---|---|---|---|---|---|---|---|---|---|
| **Method** | CC ↑ | KLD ↓ | AUC ↑ | SIM ↑ | **Method** | CC ↑ | KLD ↓ | AUC ↑ | SIM ↑ |
| ***Dataset: MIT1003 (Natural)*** | | | | | ***Dataset: U-EYE (Web page)*** | | | | |
| DVA (Wang & Shen, 2017) | 0.699 | 0.753 | 0.897 | 0.566 | SAM (Cornia et al., 2018) | 0.580 | 1.490 | 0.811 | 0.520 |
| SAM-ResNet (Cornia et al., 2018) | 0.746 | 1.247 | 0.902 | 0.597 | UMSI (Fosco et al., 2020) | 0.562 | 1.580 | 0.805 | 0.510 |
| UNISAL (Droste et al., 2020) | 0.734 | 1.014 | 0.902 | 0.597 | TransalNet (Lou et al., 2022) | 0.696 | 0.616 | 0.839 | 0.598 |
| FastSal (Hu & McGuinness, 2021) | 0.590 | 1.036 | 0.875 | 0.478 | SAM++ (Jiang et al., 2023) | 0.580 | 1.190 | 0.800 | 0.530 |
| TransalNet (Lou et al., 2022) | 0.722 | 0.660 | 0.903 | 0.592 | UMSI++ (Jiang et al., 2023) | 0.670 | 0.860 | 0.830 | 0.580 |
| SUM (Hosseini et al., 2025b) | 0.768 | 0.563 | 0.913 | 0.630 | SUM (Hosseini et al., 2025b) | 0.731 | 0.544 | 0.846 | 0.630 |
| AAM (Ours) | 0.831 | 0.446 | 0.923 | 0.674 | AAM (Ours) | 0.743 | 0.524 | 0.847 | 0.635 |
| ***Dataset: CAT2000 (Natural)*** | | | | | ***Dataset: SalECI (E-Commercial)*** | | | | |
| DVA (Wang & Shen, 2017) | 0.861 | 0.449 | 0.878 | 0.734 | SSM (Cornia et al., 2018) | 0.720 | 0.599 | 0.830 | 0.611 |
| SAM-ResNet (Cornia et al., 2018) | 0.870 | 0.670 | 0.878 | 0.739 | DeepGaze (Linardos et al., 2021) | 0.560 | 0.995 | 0.842 | 0.399 |
| MSI-Net (Kroner et al., 2020) | 0.866 | 0.428 | 0.881 | 0.730 | SSwin (Jiang et al., 2022) | 0.687 | 0.652 | 0.868 | 0.606 |
| UNISAL (Droste et al., 2020) | 0.842 | 0.530 | 0.876 | 0.721 | EML-NET (Jiang et al., 2023) | 0.510 | 1.220 | 0.807 | 0.536 |
| MDNSal (Reddy et al., 2020) | 0.889 | 0.293 | 0.878 | 0.751 | TransalNet (Jiang et al., 2023) | 0.717 | 0.873 | 0.824 | 0.534 |
| FastSal (Hu & McGuinness, 2021) | 0.721 | 0.552 | 0.860 | 0.603 | Temp-Sal (Aydemir et al., 2023) | 0.719 | 0.712 | 0.813 | 0.629 |
| TransalNet (Lou et al., 2022) | 0.877 | 0.287 | 0.882 | 0.744 | Hosseini (Hosseini et al., 2025a) | 0.750 | 0.578 | 0.892 | 0.645 |
| SUM (Hosseini et al., 2025b) | 0.882 | 0.270 | 0.888 | 0.754 | SUM (Hosseini et al., 2025b) | 0.789 | 0.473 | 0.899 | 0.680 |
| AAM (Ours) | 0.906 | 0.235 | 0.890 | 0.769 | AAM (Ours) | 0.797 | 0.450 | 0.899 | 0.678 |
| ***Dataset: SALICON (Natural)*** | | | | | ***Dataset: OSIE (Natural)*** | | | | |
| MDNSal (Reddy et al., 2020) | 0.899 | 0.217 | 0.868 | 0.797 | SAM-ResNet (Cornia et al., 2018) | 0.758 | 0.480 | 0.860 | 0.648 |
| MSI-Net (Kroner et al., 2020) | 0.899 | 0.307 | 0.865 | 0.784 | UMSI (Fosco et al., 2020) | 0.746 | 0.513 | 0.856 | 0.631 |
| UNISAL (Droste et al., 2020) | 0.879 | 0.354 | 0.864 | 0.775 | EML-NET (Jia & Bruce, 2020) | 0.717 | 0.537 | 0.854 | 0.619 |
| DeepGaze (Linardos et al., 2021) | 0.872 | 0.285 | 0.869 | 0.733 | Hosseini (Chen et al., 2023b) | 0.761 | 0.506 | 0.860 | 0.652 |
| TransalNet (Lou et al., 2022) | 0.890 | 0.220 | 0.867 | 0.783 | TransalNet (Jiang et al., 2023) | 0.791 | 0.667 | 0.923 | 0.651 |
| Temp-Sal (Aydemir et al., 2023) | 0.911 | 0.195 | 0.869 | 0.800 | UniAR (Li et al., 2024) | 0.754 | 0.547 | 0.867 | 0.647 |
| SUM (Hosseini et al., 2025b) | 0.909 | 0.192 | 0.876 | 0.804 | SUM (Hosseini et al., 2025b) | 0.861 | 0.340 | 0.924 | 0.727 |
| AAM (Ours) | 0.925 | 0.163 | 0.876 | 0.819 | AAM (Ours) | 0.901 | 0.243 | 0.933 | 0.760 |

*Table 20.* Quantitative comparison on **video** attention modeling datasets. The best results are shown in red, and the second best in blue.

| | ⊞ Video Attention Modeling | | | | | | | | | | | |
|---|---|---|---|---|---|---|---|---|---|---|---|---|
| **Method** | **DHF1K** | | | | **Hollywood2** | | | | **UCF** | | | |
| | AUC↑ | SIM↑ | CC↑ | NSS↑ | AUC↑ | SIM↑ | CC↑ | NSS↑ | AUC↑ | SIM↑ | CC↑ | NSS↑ |
| DeepVS (Jiang et al., 2018) | 0.856 | 0.256 | 0.344 | 1.911 | 0.887 | 0.356 | 0.446 | 2.313 | 0.870 | 0.321 | 0.405 | 2.089 |
| ACLNet (Wang et al., 2018) | 0.890 | 0.315 | 0.434 | 2.354 | 0.913 | 0.542 | 0.623 | 3.086 | 0.897 | 0.406 | 0.510 | 2.567 |
| TASED-Net (Min & Corso, 2019) | 0.895 | 0.361 | 0.470 | 2.667 | 0.918 | 0.507 | 0.646 | 3.302 | 0.899 | 0.469 | 0.582 | 2.920 |
| UNISAL (Droste et al., 2020) | 0.901 | 0.390 | 0.490 | 2.776 | 0.934 | 0.542 | 0.673 | 3.380 | 0.917 | 0.498 | 0.636 | 3.189 |
| HD2S (Bellitto et al., 2021) | 0.908 | 0.406 | 0.503 | 2.812 | 0.936 | 0.551 | 0.670 | 3.352 | 0.904 | 0.507 | 0.604 | 3.114 |
| ViNet (Jain et al., 2021) | 0.908 | 0.381 | 0.511 | 2.872 | 0.930 | 0.550 | 0.693 | 3.730 | 0.924 | 0.522 | 0.673 | 3.620 |
| STSANet (Wang et al., 2021) | 0.913 | 0.383 | 0.529 | 3.010 | 0.938 | 0.579 | 0.721 | 3.974 | 0.938 | 0.563 | 0.698 | 3.889 |
| ECANet (Xue et al., 2022) | 0.903 | 0.385 | 0.500 | 2.814 | 0.929 | 0.526 | 0.673 | 3.910 | 0.923 | 0.561 | 0.685 | 3.698 |
| VSSM (Lu et al., 2023) | 0.915 | 0.383 | 0.521 | 3.027 | 0.939 | 0.583 | 0.729 | 3.927 | 0.936 | 0.560 | 0.705 | 3.908 |
| TSFP-Net (Chang & Zhu, 2023) | 0.912 | 0.392 | 0.517 | 2.967 | 0.936 | 0.571 | 0.711 | 3.930 | 0.939 | 0.563 | 0.710 | 3.913 |
| MSFF-Net (Zhang et al., 2023) | 0.913 | 0.392 | 0.534 | 3.066 | 0.940 | 0.574 | 0.723 | 3.952 | 0.933 | 0.557 | 0.698 | 3.769 |
| TFS-Net (Li et al., 2025b) | 0.912 | 0.412 | 0.527 | 2.953 | 0.934 | 0.580 | 0.725 | 3.952 | 0.930 | 0.558 | 0.664 | 3.653 |
| RecSal-Net (Woo et al., 2025) | 0.913 | 0.414 | 0.547 | 3.135 | 0.938 | 0.606 | 0.737 | 3.901 | 0.918 | 0.523 | 0.644 | 3.381 |
| AAM (Ours) | 0.919 | 0.421 | 0.563 | 3.272 | 0.944 | 0.599 | 0.742 | 4.055 | 0.943 | 0.584 | 0.736 | 3.892 |

*Table 21.* Quantitative comparison on the **LEDOV** (video) dataset. The best results are shown in red, and the second best in blue.

| LEDOV | | | | |
|---|---|---|---|---|
| **Method** | AUC-J↑ | NSS↑ | CC↑ | KLD↓ |
| GBVS (Harel et al., 2006) | 0.839 | 1.541 | 0.322 | 1.824 |
| Rudoy et al. (Rudoy et al., 2013) | 0.799 | 1.454 | 0.320 | 2.421 |
| SAILICON (Huang et al., 2015) | 0.851 | 2.332 | 0.437 | 1.635 |
| OBDL (Hossein Khatoonabadi et al., 2015) | 0.801 | 1.545 | 0.315 | 2.053 |
| AWS-D (Leboran et al., 2016) | 0.795 | 1.365 | 0.294 | 2.023 |
| Xu et al. (Xu et al., 2016) | 0.827 | 1.475 | 0.382 | 1.652 |
| BMS (Zhang & Sclaroff, 2016) | 0.757 | 0.979 | 0.214 | 2.225 |
| SalGAN (Pan et al., 2017) | 0.868 | 2.193 | 0.428 | 1.680 |
| DVA (Wang & Shen, 2017) | 0.885 | 2.840 | 0.557 | 1.323 |
| DeepVS (Jiang et al., 2018) | 0.902 | 2.999 | 0.586 | 1.222 |
| ACLNet (Wang et al., 2018) | 0.897 | 2.872 | 0.570 | 1.445 |
| Sal-DCNN (Jiang et al., 2019) | 0.892 | 2.838 | 0.573 | 1.304 |
| AAM (Ours) | 0.941 | 3.652 | 0.708 | 0.888 |

*Table 22.* Quantitative comparison on **audio-visual** attention modeling datasets.

| | 🔊⊞ Audio-Video Attention Modeling | | | | | | | | | | | | | | |
|---|---|---|---|---|---|---|---|---|---|---|---|---|---|---|---|
| **Method** | **DIEM** | | | **ETMD** | | | **SumMe** | | | **Coutrot1** | | | **Coutrot2** | | |
| | CC↑ | NSS↑ | AUC↑ | CC↑ | NSS↑ | AUC↑ | CC↑ | NSS↑ | AUC↑ | CC↑ | NSS↑ | AUC↑ | CC↑ | NSS↑ | AUC↑ |
| TSFP (Chang & Zhu, 2021) | 0.651 | 2.62 | 0.906 | 0.576 | 3.07 | 0.932 | 0.464 | 2.30 | 0.894 | 0.571 | 2.73 | 0.895 | 0.743 | 5.31 | 0.959 |
| STAViS (Tsiami et al., 2020) | 0.580 | 2.29 | 0.885 | 0.585 | 3.09 | 0.934 | 0.427 | 2.10 | 0.888 | 0.497 | 2.29 | 0.868 | 0.732 | 5.64 | 0.961 |
| ViNet (Jain et al., 2021) | 0.632 | 2.53 | 0.899 | 0.571 | 3.08 | 0.928 | 0.463 | 2.41 | 0.897 | 0.560 | 2.73 | 0.889 | 0.754 | 5.95 | 0.951 |
| CASP (Xiong et al., 2023) | 0.655 | 2.61 | 0.906 | 0.620 | 3.34 | 0.940 | 0.499 | 2.60 | 0.907 | 0.561 | 2.65 | 0.889 | 0.788 | 6.34 | 0.963 |
| DAVS (Zhu et al., 2024) | 0.580 | 2.29 | 0.884 | 0.600 | 2.96 | 0.932 | 0.423 | 2.29 | 0.889 | 0.482 | 2.19 | 0.869 | 0.734 | 4.98 | 0.960 |
| MSPI (Xie et al., 2024) | 0.653 | 2.62 | 0.907 | 0.601 | 3.24 | 0.937 | 0.482 | 2.49 | 0.901 | 0.567 | 2.76 | 0.895 | 0.783 | 6.28 | 0.963 |
| TAVDiff (Yu et al., 2025) | 0.670 | 2.75 | 0.909 | 0.613 | 3.15 | 0.937 | 0.500 | 2.51 | 0.904 | 0.607 | 2.85 | 0.892 | 0.798 | 6.52 | 0.963 |
| AAM (Ours) | 0.710 | 2.88 | 0.919 | 0.655 | 3.66 | 0.945 | 0.550 | 2.90 | 0.920 | 0.626 | 3.22 | 0.911 | 0.887 | 7.46 | 0.971 |

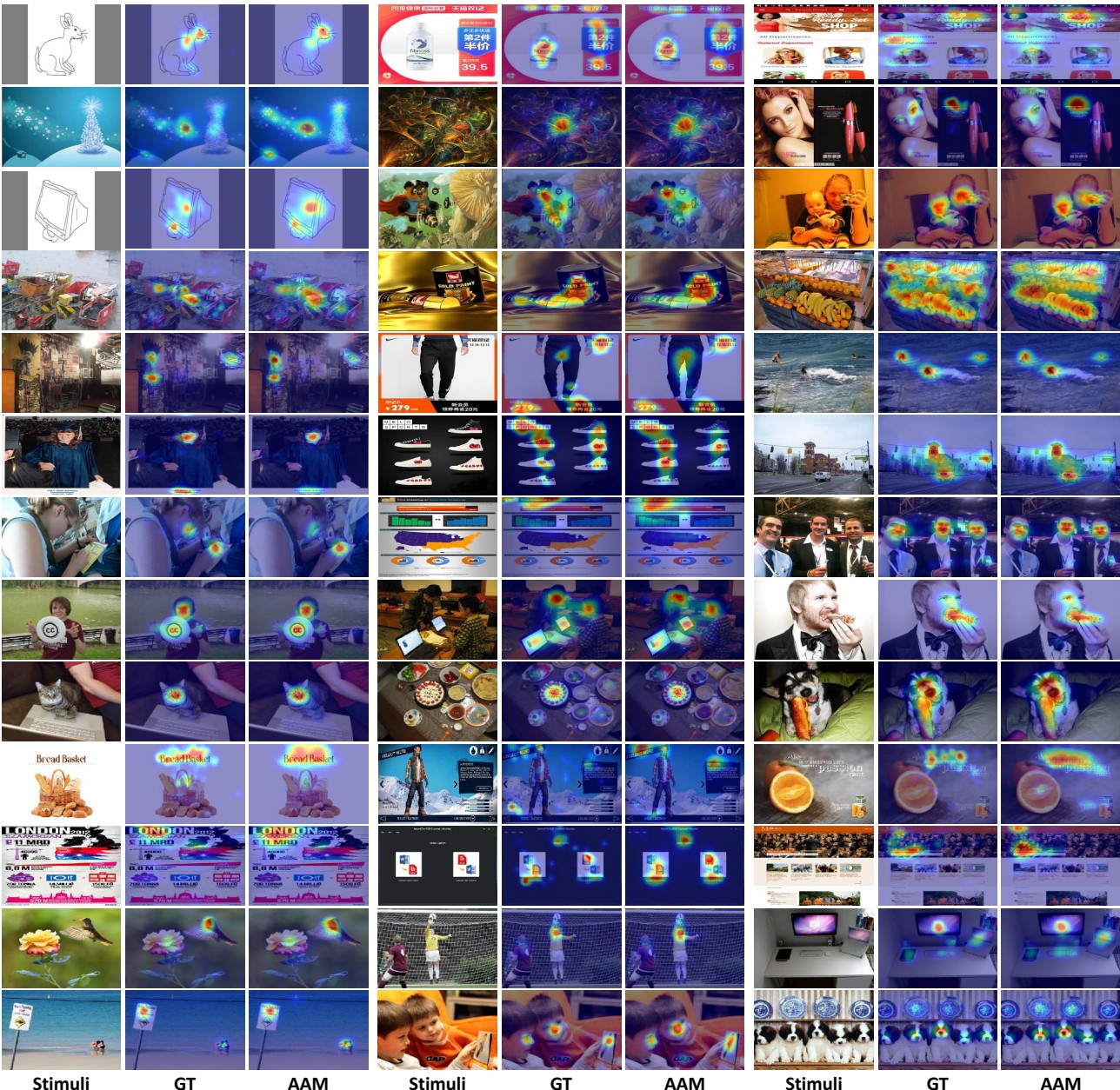

*Figure 16.* Prediction results of AAM (Ours) across various task scenarios, where GT denotes the human attention distribution.

