# OpenReview forum: "Attend to Anything: Foundation Model for Unified Human Attention Modeling"
_ICML.cc/2026/Conference — ICML 2026 regular_

### Official Review · Reviewer_QFr4 · 2026-03-09

**Soundness:** 3
**Presentation:** 3
**Significance:** 3
**Originality:** 3
**Overall Recommendation:** 4
**Confidence:** 3

**Summary:**

This paper proposes AAM, a unified model for human attention prediction across images, videos, and audio-visual inputs. The method combines prompt-conditioned hyperbolic representations for hierarchical attention modeling with a Fokker–Planck-inspired temporal dynamics module for video attention. The paper also introduces a unified training corpus, Attention-1.75M, and reports strong results on 16 benchmarks with improvements over prior methods and faster video inference.

**Compliance With Llm Reviewing Policy:**

Affirmed.

**Final Justification:**

The rebuttal adequately addresses my concerns. The commitment to detailing dataset biases and potential misuse is also appreciated. I have no further questions and maintain my recommendation of Weak Accept.

**Key Questions For Authors:**

How much of the gain comes from hyperbolic geometry versus prompt conditioning alone? A matched Euclidean prompt-conditioned baseline would help.

Can the authors provide stronger cross-dataset or out-of-domain generalization results to support the unification claim?

How sensitive is performance to the design of the dataset-level prompts?

Are the reported speed comparisons done under fully matched settings?

**Limitations:**

No. The paper should discuss dataset bias, prompt bias, and possible misuse in persuasive design or surveillance more explicitly.

**Strengths And Weaknesses:**

Strengths:
The paper addresses an important problem, namely the fragmentation of saliency modeling across modality and task. The overall framework is ambitious and coherent, and the combination of hyperbolic hierarchy modeling and temporal dynamics is interesting. Experimental coverage is broad, and the efficiency gains for video inference are promising. The prompt-based analyses are also useful.

Weaknesses:
The “foundation model” framing feels somewhat stronger than what is fully demonstrated, since the training data is mainly a curated union of existing datasets with dataset-level prompts. It is also not fully clear how much of the gain comes from hyperbolic geometry itself versus prompt conditioning, joint training, or decoder design. The PDE interpretation of the temporal module is appealing but only partially justified. Finally, reproducibility would be stronger with clearer release details for code, prompts, and the unified dataset.

---

> ### Author Rebuttal · Authors · 2026-03-29
>
> # [Q1/Weak1] Hyperbolic Geometry vs. Prompt Conditioning Gains:
>
> We have included a matched Euclidean prompt-conditioned joint training as detailed in **Sec. 4.3.2 (Figs. 5–6, F1: Euclidean Prompt) and Appendix E.1 (Table 10)**. These results provide detailed ablations comparing Euclidean and hyperbolic spaces under identical prompt conditioning.
>
> Furthermore, in Sec. 4.3 and Appendix E.1 (Tables 11–13), we conduct comprehensive ablations to disentangle the contributions of **hyperbolic geometry, prompt conditioning, joint training, and decoder design**. Fig. 7 visualizes the learned hyperbolic representations, illustrating that AAM effectively captures the hierarchical structure of attention.
>
> **Attribution of Performance Gains:** The results suggest that **prompt-only joint training may lead to performance degradation**, whereas AAM’s performance is supported by rethinking human attention as a unified hierarchical cognitive process.
>
> # [Q2] Generalization Results
> We have provided the zero-shot generalization results for AAM on two datasets in **Section 4.3.2 and Appendix E.3 (Table 14).** To further substantiate the cross-domain generalization of AAM, we include **additional zero-shot performance** evaluations across four heterogeneous datasets. All reported values are measured using the CC metric.
>
> |*Image Datasets*|COCO|Toronto|*Video Datasets*|ECCV-AIM|LEDOV|*Audio-Visual Datasets*|CVPR-NTIRE|AVAD|
> |-|-|-|-|-|-|-|-|-|
> |*EML-NET*|0.68|0.71|*DeepVS*|0.62|0.59|*NTR*|0.69|-|
> |*UNISAL*|0.71|0.81|*ACLNet*|0.65|0.57|*MSPI*|-|0.70|
> |*Ours (Zero-shot)*|0.71|0.82|*Ours (Zero-shot)*|0.65|0.71|*Ours (Zero-shot)*|0.69|0.71|
>
> # [Q3] Analysis of Prompt Sensitivity:
>
> We respectfully clarify that AAM is **robust to prompt variations**, as demonstrated in **Sec. 4.3.3 (Fig. 6) and Appendix E.2 (Table 15, Figs. 14–15)**.
> Specifically, we evaluate AAM under the following controlled settings:
>
> - **(1) Condition swap** (incorrect/null prompts)
> - **(2) Paraphrase invariance** (synonymous prompt variations)
> - **(3) Prompt granularity** (progressively ambiguous)
>
> AAM exhibits **no significant performance sensitivity** under synonymous rewriting, and **maintains strong performance** with only minor degradation even when subjected to incorrect prompts.
>
> # [Q4] Fairness of Speed Comparisons:
>
> All speed comparisons reported in this paper were conducted under **strictly controlled and identical settings** to ensure fairness.
>
> - **Hardware Consistency:** All models were evaluated on a single NVIDIA A100 GPU.
>
> - **Evaluation Protocol:** For existing video methods, we followed the standard sliding window paradigm, which takes 32 frames as input to predict the saliency map of the last frame. In contrast, AAM generates predictions for all frames in a single forward pass via the FPD module, inherently achieving higher throughput while maintaining competitive performance.
>
> # [Weak2] PDE interpretation of FPD
>
> We clarify that the theoretical justification based on a continuous PDE formulation is provided in Appendix C (Theoretical Proof and Analysis).  To connect the discrete neural implementation with the PDE formulation, we employ a **Lie–Trotter operator splitting scheme** (Sec. 3.4 and App. C.1):
>
> - **Mapping:** As derived in App. C.2, the drift term (∂u / ∂t = −∇⋅(vu)) is parameterized as a valid Markov kernel.  The diffusion and assimilation steps are implemented as corresponding discrete updates.
>
> - **Physical Constraints:** In Proposition C.1, we show that the module preserves mass conservation (simplex invariance) and maintains numerical stability.  By bounding the diffusion step size via the CFL condition (Δtν), we ensure non-negativity and avoid oscillatory behavior.
>
> We will add a forward reference in Sec. 3 to guide readers to Appendix C.
>
> # [Weak3] Data/Code:
>
> We are committed to **releasing the demo, source code, and datasets** upon acceptance. Furthermore, we have provided comprehensive and **reproducible details in Appendix B, Figs. 8-10, and Figs. 14-15**. Specifically, the pseudo-code and architectural diagrams for the decoder and audio modules are included in Appendix D (Methodology Details) to facilitate implementation.
>
> # [Limitation] Biases and Potential Misuses:
>
> We thank the reviewer for highlighting these societal impacts. We will add a "Broader Impacts" section explicitly discussing:
>
> - **Inherent Biases**: We provide summary statistics of Attention-1.75M; more detailed quantitative analysis will be presented in Appendix B.
> - **Potential Misuses**: We explicitly warn against exploiting AAM for "persuasive design" (e.g., engineering addictive interfaces or dark patterns to manipulate user attention).

---

> > ### Author Rebuttal · Reviewer_QFr4 · 2026-04-02
> >
> > The rebuttal adequately addresses my concerns. The commitment to detailing dataset biases and potential misuse is also appreciated. I have no further questions and maintain my recommendation of Weak Accept.

---

> > > ### Author Response · Authors · 2026-04-02
> > >
> > > We sincerely thank the reviewer for their time, and valuable suggestions, all of which have greatly contributed to improving the quality of this work. We are glad that our clarifications have helped resolve the key questions raised, including the contributions of hyperbolic geometry, prompt robustness, generalization ability.
> > >
> > > We hope that these revisions provide a more complete and reliable understanding of our work, and further strengthen confidence in its technical soundness and practical value.

---

### Official Review · Reviewer_w9Kw · 2026-03-11

**Soundness:** 3
**Presentation:** 4
**Significance:** 3
**Originality:** 3
**Overall Recommendation:** 5
**Confidence:** 4

**Summary:**

This paper introduces the Attend to Anything Model (AAM), a unified multimodal foundation model designed to predict human visual attention across diverse data modalities, including images, videos, and audio-visual streams. The core contribution is a task-conditioned framework that utilizes attention task description prompts to learn multiple saliency tasks jointly within a single architecture.

The authors demonstrate the scalability and efficiency of AAM through extensive evaluation on 16 benchmarks. The model consistently outperforms existing state-of-the-art methods by an average margin of 6%, while notably achieving a 4x speedup in video inference. This suggests a significant improvement in both the generalizability of attention modeling and its computational efficiency for real-time applications.

**Compliance With Llm Reviewing Policy:**

Affirmed.

**Ethical Review Concerns:**

The curated Attention-1.75M benchmark aggregates 18 existing datasets. The final paper should provide a clear Data Provenance and Licensing Matrix. Specifically, the paper must explicitly detail the original terms of use for each of the 18 constituent datasets and specify the licensing framework under which the new aggregate benchmark will be distributed to the community (e.g., CC BY 4.0, Research Only, etc.).

**Ethical Review Flag:**

Flag this paper for an ethics review.

**Ethics Expertise Needed:**

["Legal Compliance (e.g., EU AI Act, GDPR, copyright, terms of use)"]

**Final Justification:**

The rebuttal adequately addresses my concerns. My recommendation is remain the same.

**Key Questions For Authors:**

- Can the author confirm that the Attention-1.75M dataset is a combination of existing dataset with  attention collection details (e.g., domain and task) being described and used as a text prompt? It would be good to clarify what are the additional contribution in standardising this benchmark.

- The Attention-1.75M benchmark, which aggregates 18 existing datasets to provide a unified platform for human attention modeling. However, given the scale and the multi-source nature of this corpus, it is imperative that the authors provide a clear Data Provenance and Licensing Matrix. Specifically, the paper must explicitly detail the original terms of use for each of the 18 constituent datasets and specify the licensing framework under which the new aggregate benchmark will be distributed to the community (e.g., CC BY 4.0, Research Only, etc.). Ensuring that the benchmark complies with the restrictive or permissive licenses of its components is vital for the long-term legal and ethical viability of this foundation model. This transparency is necessary to confirm that the work adheres to the data usage policies of the original authors and the conference standards.

- The paper claims that AAM introduces a "cognitively aligned paradigm." Beyond empirical performance on benchmarks, what specific theoretical evidence or psychological principles support this alignment? For instance, does the Hyperbolic Space mapping align with known hierarchical processing in the visual cortex (e.g., the V1-V4 hierarchy)? Furthermore, does the Fokker-Planck Dynamic Module (FPD) relate to established models of human visual sampling, such as "Stochastic Drift-Diffusion Models" (DDM) often used in cognitive science to explain decision-making and saccadic latency? Providing a formal link between these mathematical choices and biological attention mechanisms would significantly strengthen the paper’s positioning.

**Limitations:**

Yes

**Strengths And Weaknesses:**

Strengths
- Unification of Fragmented Tasks: A major strength of this work is the transition from modality-specific attention models to a unified foundation model. By consolidating human attention modeling across images, videos, and audio-visual streams, the authors address the persistent issue of poor cross-scene generalization in the field.

- Technical Contribution: The use of Hyperbolic Space to reformulate attention is technically sound and intuitive. Because hyperbolic geometry naturally supports hierarchical structures via its exponential expansion of volume, it provides a principled geometric constraint for modeling the "general-to-specific" nature of human saliency across different scenarios. The introduction of the Fokker-Planck Dynamic Module (FPD) is a novel and efficient way to handle video attention. By treating temporal evolution as a diffusive process governed by the Fokker-Planck equation, the model bypassing the computational redundancy of traditional multi-frame fusion or 3D-CNN approaches. This frame-wise efficiency is a significant contribution to real-time video saliency.

- Standardise Benchmark: The creation of Attention-1.75M, by combining18 datasets from the literature, is an empirical contribution. Providing a standardized corpus with over 1.7 million instances—spanning diverse domains from e-commerce to cinematic content—will likely serve as a foundational benchmark for future multimodal attention research. Despite that, I strongly tribute the contributions to various authors that take enormous effort in collecting the base dataset that are now included as part of this benchmark. Their contribution should not be lightly ignore.

Weaknesses
- Sensitivity to Prompt Engineering: While the task-conditioned nature of the AAM model is a strength, it introduces a significant dependency on textual prompt quality. The review notes that "wrong" or mismatched prompts lead to a strict performance hierarchy, suggesting the model may not be as robust to natural language variations as a true "foundation model" should be.

- Lack of Principle-Driven Prompting: Despite the goal of moving toward a general-purpose attention model, it still rely on dataset-level prompts derived from original acquisition protocols. The work would be significantly strengthened if the prompts were derived from first principles of human attention modeling (e.g., bottom-up vs. top-down saliency, foveal vs. peripheral vision) rather than just metadata-based descriptions. This would bridge the gap between computer vision performance and the psychological reality of human attention.



Overall, this paper provide unique insights to unify human attention modelling with strong empirical results. The formulation of attention as a hierarchical entailment process and model temporal attention evolution with FPD is a good contributions.
The proposed AAM requires significant computational resources for training. While this is expected for a foundation model of this scale, investigating more efficient training regimes or distillation strategies remains a vital direction for future research.
The Attention-1.75M corpus is a major asset for the community. However, it is essential that the authors provide explicit licensing details for both the aggregate corpus and the underlying datasets used in its creation to ensure compliance and promote open-source adoption.

---

> ### Author Rebuttal · Authors · 2026-03-27
>
> # [Q1/Weak2] Contributions and Cognitive Prompts of Attention-1.75M:
>
> Our work is not positioned as a direct counterpart to dataset-focused studies. However, it still contributes meaningfully at the dataset level. In particular, this standardization serves as a **foundational step toward rethinking attention modeling**. We will explicitly include these in Appendix B:
>
> - **1) Cognitive-Driven Formulation:** We reorganize tasks along the **first principles of human attention hierarchy** (from bottom-up stimuli to top-down goals). Encoding these semantics into prompts transforms isolated datasets into a structured semantic space:
>
> |Level|Attention|Description|Template|Datasets|
> |-|-|-|-|-|
> |High|**Top-down**|Guided by explicit goals, task objectives| task setting + bias|Hollywood, UCF, SumMe|
> |Mid|**Semantic Modulation**|Influenced by objects, interactions and semantic relationships|key content elements|OSIE, FIWI, UI, SalEC, DIEM, AVAD, Coutrot_db, ETMD_av|
> |Low|**Bottom-up**|Driven by stimuli such as contrast, motion, scene statistics|stimulus + domain|MIT1003, CAT2000, SALICON, DHF1K|
>
> - **2) Unified Paradigm:** The Attention-1.75M benchmark unifies previously fragmented tasks into a single formulation for cross-domain training. This unified setup enables capabilities such as cross-task generalization and zero-shot reasoning, which are difficult to achieve with conventional pipelines.
>
> # [Q2/Ethical Concerns] Data Provenance and Licensing Matrix:
>
> We have carefully reviewed the terms of use for all 18 datasets and **will include a detailed Data Provenance and Licensing Matrix** in Appendix B, documenting the original license, usage restrictions, and redistribution permissions for each dataset.
>
> - **CC BY 4.0**: SALICON, U-EYE, DHF1K, Hollywood-2, UCF.
> - **Research Only** (following original publications): MIT300, CAT2000, FIGRIM, FiWI, LEDOV, Coutrot_db, DIEM, ETMD, SumMe.
> - **MIT License**: OSIE, SalECI.
>
> **Aggregate Benchmark.** To ensure compliance with the most restrictive components, Attention-1.75M will be released under a **research-only** policy. We do not redistribute datasets with restricted licenses. Instead, we provide: (1) standardized preprocessing pipelines; and (2) direct links to the original sources.
>
> # [Q3] Theoretical Basis for AAM:
> Existing biological studies provide strong theoretical support for AAM, bridging the gap between machine learning and cognitive science. AAM provides a cognitively aligned formulation for both the **static spatial geometry and the temporal dynamic process of biological attention**, which we **will further elaborate** in Section 2.
>
> - **1) Hyperbolic Space $\iff$ Visual Cortex Hierarchy**
>
> **Bio-Basis:** The visual cortex processes information via a hierarchy (from simple edges in V1 to complex semantics in IT). Prior studies demonstrate that visual hypercolumns are natively modeled as hyperbolic (Chossat & Faugeras, 2009), and actual spiking patterns in V1/V2 exhibit an inherent hyperbolic geometry (Guidolin et al., 2022).
>
> **Link:** AAM's cognitively aligned hyperbolic space mathematically mirrors the **foundational geometry of biological vision**.
>
> - **2) FPD $\iff$ Stochastic Drift-Diffusion Model (DDM)**
>
> **Bio-Basis:** In cognitive science, the DDM is the established standard for explaining perceptual decision-making and saccadic latency (Ratcliff, 1978; Bogacz et al., 2006). Mathematically, the Fokker-Planck equation is the exact analytical counterpart governing the macroscopic probability density of such a DDM process.
>
> **Link:** FPD provides a mechanistic account of **attention shifts grounded in biological evidence accumulation**. The "drift" mathematically formulates the top-down task-driven bias, while "diffusion" captures bottom-up visual exploration. The concentration of probability mass in FPD triggers an attention shift, directly analogous to decision boundary crossing in DDM (Shinn et al., 2020).
>
>
>
> # [Weak1] Sensitivity to Prompt Quality and Engineering:
>
> As we have detailed in **Sec. 4.3.3 (Fig. 6) and Appendix E.2 (Table 15, Figs. 14–15)**, AAM is **robust to natural language variations** as a "foundation model." Specifically, we evaluate AAM under the following controlled settings:
>
> - **(1) Condition swap (incorrect/null prompts)**
> - **(2) Paraphrase invariance (synonymous prompt variations)**
> - **(3) Prompt granularity (progressively ambiguous)**
>
> (2) Paraphrase e.g.:
> - "No explicit goal is given. Predict saliency under natural viewing."
> - "Predict general visual attention for free-viewing."
>
> In Table 15, AAM exhibits no significant performance variance under synonymous rewriting, and **maintains strong performance** with only minor degradation even when subjected to incorrect prompts. Importantly, the slight sensitivity to condition prompts highlights the role of AAM’s hierarchy in achieving a favorable **trade-off** between genuine architectural robustness and hierarchical attention mechanisms.

---

> > ### Author Rebuttal · Reviewer_w9Kw · 2026-04-01
> >
> > The rebuttal has addressed my concerns. I will keep my original rating.

---

> > > ### Author Response · Authors · 2026-04-02
> > >
> > > We sincerely thank the reviewer for the positive and encouraging evaluation. We particularly appreciate the recognition of our cognitively inspired unified modeling of attention.
> > >
> > > Following these suggestions, we will further clarify the cognitive grounding of our framework to explicitly strengthen the connection between our mathematical formulation and biological attention mechanisms. We will also include a detailed data provenance and licensing matrix to ensure transparency and compliance.
> > >
> > > We believe these additions will further enhance the clarity, reliability, and long-term impact of our proposed benchmark and model within the attention research community.

---

### Official Review · Reviewer_B9At · 2026-03-12

**Soundness:** 2
**Presentation:** 2
**Significance:** 3
**Originality:** 3
**Overall Recommendation:** 4
**Confidence:** 3

**Summary:**

This paper proposes a novel and highly influential foundation model (AAM) for unified human attention modeling across image, video, and audio-visual modalities, addressing the long-standing issue of research fragmentation in this field. The work is characterized by both theoretical rigor and methodological innovation. Extensive empirical experiments were conducted on 16 benchmark datasets, demonstrating that the proposed AAM model not only achieves state-of-the-art performance but also significant efficiency improvements. Furthermore, its cognitively inspired design principles offer valuable theoretical insights for holistic visual perception research.

**Compliance With Llm Reviewing Policy:**

Affirmed.

**Final Justification:**

The rebuttal adequately addresses my concerns. My recommendation is remain weak accept.

**Key Questions For Authors:**

1. AAM claims a 4x video inference speedup (Table 4). Please provide a detailed breakdown of the computational cost of hyperbolic operations versus equivalent Euclidean implementations. Crucially, clarify whether this speedup is primarily due to the FPD module or specific optimizations in the hyperbolic implementation.
2. Regarding the "dataset-level prompts" in Attention-1.75M (Appendix B), please detail the annotation quality control and consistency measures. Furthermore, provide a quantitative analysis of potential biases (e.g., scene, content, demographic) within the dataset.
3. The model's response to varied prompts is shown in Inductive Analysis (Section 4.3.3). Please present empirical evidence or analysis on AAM's robustness against 'incorrect' or 'ambiguous' prompts. Are there experiments demonstrating resilience against catastrophic failure, or proposed strategies to enhance robustness?
4. To better quantify the contribution of hyperbolic geometry, please provide comparative ablation results for: (a) removing the HAE loss, and (b) modeling solely in Euclidean space.

**Limitations:**

yes

**Strengths And Weaknesses:**

Strengths:

1.The model effectively integrates hyperbolic geometry and the Fokker–Planck equation, both of which are well-established mathematical tools with robust theoretical foundations in their respective domains. Applying these to attention modeling provides a strong theoretical underpinning for the proposed method.

2.The reframing of attention modeling as a hierarchical entailment relationship, learned through geometric constraints in hyperbolic space, is a highly creative perspective. The idea of using Fokker–Planck dynamics to unify video attention is also quite ingenious.

3.Standard metrics such as AUC-Judd (AUC), Similarity Metric (SIM), Linear Correlation Coefficient (CC), and Normalized Scanpath Saliency (NSS) have been employed. Furthermore, the ablation studies thoroughly validate the effectiveness of individual components.

4.The authors demonstrate a rigorous approach to their research by discussing the results of ablation studies and providing inductive analyses of the model's behavior, which indicates a commitment to honest evaluation.

Weaknesses:

1. The discrete implementation of Fokker–Planck dynamics (FPD) (e.g., Lie–Trotter) requires more detailed empirical evidence on numerical stability in complex video scenarios, especially regarding its reliance on the CFL condition.

2. Analysis of the computational overhead of hyperbolic geometry operations is insufficient, lacking direct comparison with Euclidean counterparts. The attribution of speedup (FPD vs. hyperbolic optimization) needs clarification.

3. The annotation quality and generation process for "dataset-level prompts" in the Attention-1.75M dataset, as well as a quantitative analysis of potential biases, are not sufficiently detailed.

---

> ### Author Rebuttal · Authors · 2026-03-26
>
> # [Q1/Weak2] Hyperbolic Cost and Speedup Attribution:
>
> ||FLOPs (M)|Time (ms)|U-EYE (CC)|
> |-|-|-|-|
> |Euclidean|0.197|8.8|0.63|
> |Hyperbolic|0.201|12.6|0.74|
>
> We replace the hyperbolic embedding operations with Euclidean space under the same setting (500 samples). The hyperbolic setting only incurs **negligible additional computational overhead** overhead while providing significant gains. We will include additional analysis in Sec. 4.3.2.
>
> **The reported 4× speedup is attributed to the FPD**. As analyzed in Section 4.2, existing methods rely on a sliding-window strategy (32-frame input→**1-frame output**), processing a 32-frame video 32 times, which leads to redundant computations.. In contrast, AAM employs a frame-wise prediction via FPD (32-frame input → **32-frame output**) in a single forward pass. Moreover, after removing the temporal downsampling operations in existing methods, applying FPD yields over a 7× speedup.
>
> # [Q2/Weak3] Prompt Quality Control and Dataset Analysis:
> We will supplement the following details in Appendix B:
>
> - **Annotation**:
> i) We reorganize tasks along the **first principles of human attention hierarchy**:
>
> |Level|Attention|Description|Template|Datasets|
> |-|-|-|-|-|
> |High|**Top-down**|Guided by explicit goals and task objectives|task setting + bias|Hollywood, UCF, SumMe|
> |Mid|**Semantic Modulation**|Influenced by objects, interactions and semantic relationships|key content elements|OSIE, FIWI, UI, SalEC, DIEM, AVAD, Coutrot_db, ETMD_av|
> |Low|**Bottom-up**|Driven by visual stimuli such as contrast, motion, and scene statistics|stimulus type + domain|MIT1003, CAT2000, SALICON, DHF1K|
>
> Building upon this hierarchy, we construct prompts using the **standardized template:**
>
> **[stimulus type] + [scene/domain] + [key elements] + [task setting] + [attention bias]**
>
> ii) Prompts are further reviewed through a two-stage process (independent annotation followed by consensus adjudication) to improve consistency.
>
> - **Analysis of Dataset**: We provide summary statistics of Attention-1.75M; more detailed quantitative analysis will be presented in Appendix B.
>
> |Aspect|Quantitative Bias|
> |-|-|
> |Modality| Static images dominate (88.0%), while video (11.3%) and audio-visual (0.7%) data are scarce|
> |Image Scene| Natural scenes account for 88.3% (>70% landscape-style), while UI/web/e-commerce data contribute 11.7%|
> |Video Scene |Professional media (49.6%) and daily videos (43.0%) dominate, with limited sports (4.1%) and meetings (3.3%) |
> |Content|35% gaze on faces and 10% on bodies; text regions attract ≈5× higher fixation density in UI data|
> |Demographic/Protocol|Aged 18–36 (≈65–70% in 18–25; <1% above 36; 0% <18 or >55); >50% annotations rely on mouse-tracking proxies|
>
> # [Q3] Prompt Robustness and Failure Resilience:
>
> - **Robustness**: We have provided that AAM is robust to prompt variations in **Sec. 4.3.3 (Fig. 6) and Appendix E.2 (Table 15, Figs. 14, 15)**. Specifically, **incorrect prompts (Condition Swap)** and G1-G4 (Prompt Granularity) represent varying levels of **prompt ambiguity**. Figures 14, 15 present reproducible experimental settings.
>
> |Prompt Setting|CAT2000 (CC)|SalECI (CC)|
> |-|-|-|
> |Correct (G4)|0.90|0.79|
> |Other Method|0.84|0.75|
> |Incorrect|0.85|0.74|
> |G3 (Sub-ambiguous)|0.87|0.77|
> |G1 (Most-ambiguous)|0.86|0.73|
>
> - **Failure Resilience**: As shown in the partial results in **Table 15** , even under incorrect or highly ambiguous (G1) prompts, AAM maintains strong performance with only moderate degradation, outperforming the prior methods. While slight performance drops are observed on rare datasets, no severe degradation is observed. This highlights the role of AAM’s hierarchical attention in achieving a favorable **trade-off** between genuine robustness and hierarchical attention cognition.
>
> # [Q4] Ablation Study on Hyperbolic Contributions:
>
> We have provided comparative results in **Section 4.3.2, (F)**. Please check Appendix E.2 (Table 10) for detailed results. We additionally include an ablation without HAE loss:
>
> ||MIT1003 (CC)|CAT2000 (CC)|
> |-|-|-|
> |Solely Euclidean Space|0.77|0.82|
> |w/o Hyp Decoder|0.79|0.88|
> |w/o HAE Loss|0.79|0.86|
> |AAM|0.82|0.91|
>
> Additionally, PCA and CO-SNE visualizations of the Euclidean space will be included in Appendix E..
>
> # [Weak1] FPD Numerical Stability:
>
> FPD is designed with mechanisms that promote numerical stability (please see **Appendix C**):
>
> - **Bounded Rates:** Sigmoid gating and explicit limits (fp_alpha_max = 0.5) constrain drift and diffusion magnitudes, which helps control step sizes.
>
> - **Mass Conservation:** Clamping and ε-normalization are used to maintain non-negativity and avoid numerical instability (e.g., division by zero).
>
> - **Gradient Stabilization:** Detaching drift gradients (fp_detach_A = True) and residual gating help stabilize optimization and prevent feature collapse.

---

> > ### Author Rebuttal · Reviewer_B9At · 2026-04-01
> >
> > This addresses my concern; however, I am unlikely to revise my score, as I believe it is already sufficiently high.

---

> > > ### Author Response · Authors · 2026-04-02
> > >
> > > We sincerely thank reviewer for the insightful comments on hyperbolic computational cost, and dataset construction. These suggestions have significantly improved the technical clarity and completeness of our work.
> > >
> > > We have added detailed empirical analyses on hyperbolic cost, and expanded the discussion on annotation quality and potential biases. We are glad that our responses have addressed the reviewer’s concerns, and we hope the strengthened analysis further reinforces the soundness and impact of the paper.
> > >
> > > Once again, we thank the reviewer for their time, and valuable suggestions, all of which have greatly contributed to improving the quality of this work. We are actively available should any further questions or clarifications arise.

---

### Decision · Program_Chairs · 2026-04-30

**Decision:**

Accept (regular)

**Comment:**

The paper initially received reviews of 544 (Accept, 2x Weak Accept).  The main concerns were:

1) discrete implementation of FPD could be numerically unstable [B9At]
2) missing analysis of computational overhead [B9At, QFr4]
3) needs more details about annotation quality and biases of the new dataset [B9At]
4) sensitivity to prompts [w9Kw]
5) dataset-driven prompts [w9Kw, QFr4]
6) Data Provenance and Licensing Matrix is needed [w9Kw]
7) how is AAM a "cognitively aligned paradigm"? [w9Kw]
8) missing ablatoin study on hyperbolic geometry vs. prompt conditioning [QFr4]
9) cross-dataset or OOD generalization to support unification claim? [QFr4]

The authors wrote a reponse to address these concerns, and all reviewers were satisfied with the response and maintained their positiive ratings. Overall, the reviewers appreciated the unified dataset and modeling approach. The authors are suggested to revise the paper according to the reviews and rebuttal for the camera-ready version.